# Scalable One-Pass Optimisation of High-Dimensional Weight-Update Hyperparameters by Implicit Differentiation

**Ross M. Clarke**
University of Cambridge
rmc78@cam.ac.uk

**Elre T. Oldewage**
University of Cambridge
etv21@cam.ac.uk

**José Miguel Hernández-Lobato**
University of Cambridge
Alan Turing Institute
jmh233@cam.ac.uk

## Abstract

Machine learning training methods depend plentifully and intricately on hyperparameters, motivating automated strategies for their optimisation. Many existing algorithms restart training for each new hyperparameter choice, at considerable computational cost. Some hypergradient-based one-pass methods exist, but these either cannot be applied to arbitrary optimiser hyperparameters (such as learning rates and momenta) or take several times longer to train than their base models. We extend these existing methods to develop an approximate hypergradient-based hyperparameter optimiser which is applicable to any continuous hyperparameter appearing in a differentiable model weight update, yet requires only one training episode, with no restarts. We also provide a motivating argument for convergence to the true hypergradient, and perform tractable gradient-based optimisation of independent learning rates for each model parameter. Our method performs competitively from varied random hyperparameter initialisations on several UCI datasets and Fashion-MNIST (using a one-layer MLP), Penn Treebank (using an LSTM) and CIFAR-10 (using a ResNet-18), in time only 2–3x greater than vanilla training.

## 1 Introduction

Many machine learning methods are governed by *hyperparameters*: quantities other than model *parameters* or *weights* which nonetheless influence training (e.g. optimiser settings, dropout probabilities and dataset configurations). As suitable hyperparameter selection is crucial to system performance (e.g. Kohavi & John (1995)), it is a pillar of efforts to automate machine learning (Hutter et al., 2018, Chapter 1), spawning several hyperparameter optimisation (HPO) algorithms (e.g. Bergstra & Bengio (2012); Snoek et al. (2012; 2015); Falkner et al. (2018)). However, HPO is computationally intensive and random search is an unexpectedly strong (but beatable; Turner et al. (2021)) baseline; beyond random or grid searches, HPO is relatively underused in research (Bouthillier & Varoquaux, 2020).

Recently, Lorraine et al. (2020) used gradient-based updates to adjust hyperparameters *during* training, displaying impressive optimisation performance and scalability to high-dimensional hyperparameters. Despite their computational efficiency (since updates occur before final training performance is known), Lorraine et al.'s algorithm only applies to hyperparameters on which the loss function depends explicitly (such as $\ell_2$ regularisation), notably excluding optimiser hyperparameters.

Our work extends Lorraine et al.'s algorithm to support arbitrary continuous inputs to a differentiable weight update formula, including learning rates and momentum factors. We demonstrate our algorithm handles a range of hyperparameter initialisations and datasets, improving test loss after a single training episode ('one pass'). Relaxing differentiation-through-optimisation (Domke, 2012) and hypergradient descent's (Baydin et al., 2018) exactness allows us to improve computational and memory efficiency. Our scalable one-pass method improves performance from arbitrary hyperparameter initialisations, and could be augmented with a further search over those initialisations if desired.

## 2 Weight-Update Hyperparameter Tuning

In this section, we develop our method. Expanded derivations and a summary of differences from Lorraine et al. (2020) are given in Appendix C.

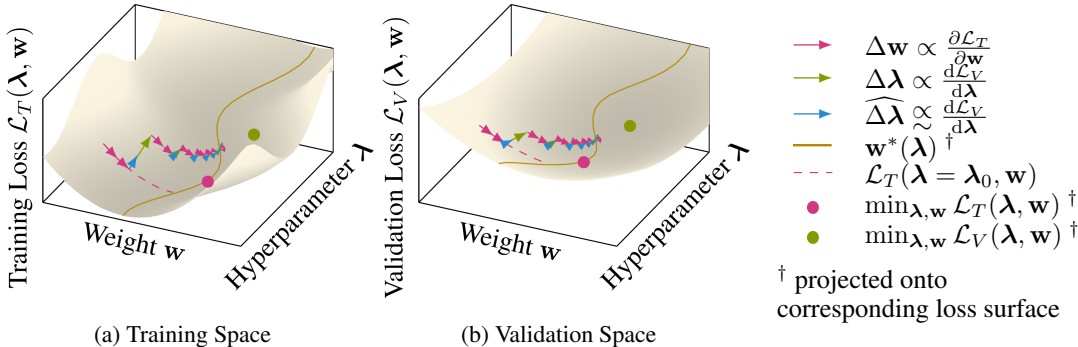

(a) Training Space          (b) Validation Space

Figure 1: Summary of our derivation (Section 2): an example of online hypergradient descent using exact hypergradients from the implicit function theorem (IFT) (→), with our method's approximate hypergradients (→) superimposed. We target optimal validation loss (●), adjusting weights $\mathbf{w}$ based on the training loss. Classical weight updates (for fixed hyperparameters $\boldsymbol{\lambda}$) converge (→, - - -) to the best-response line $\mathbf{w}^*(\boldsymbol{\lambda})$ (—); the IFT gives hyperparameter updates (→) leading to a minimum of validation loss along $\mathbf{w}^*(\boldsymbol{\lambda})$. Our approximate hyperparameter updates (→) differ in magnitude from these exact updates, but still give useful guidance.

## 2.1 IMPLICIT FUNCTION THEOREM IN BILEVEL OPTIMISATION

Consider some model with learnable parameters $\mathbf{w}$, training loss $\mathcal{L}_T$, optimisation hyperparameters $\boldsymbol{\lambda}$ and validation loss $\mathcal{L}_V$. We use $\mathbf{w}^*, \boldsymbol{\lambda}^*$ to represent the optimal values of these quantities, found by solving the following *bilevel optimisation problem*:

$$^{(a)} \boldsymbol{\lambda}^* = \arg\min_{\boldsymbol{\lambda}} \mathcal{L}_V(\boldsymbol{\lambda}, \mathbf{w}^*(\boldsymbol{\lambda})), \qquad \text{such that} \qquad ^{(b)} \mathbf{w}^*(\boldsymbol{\lambda}) = \arg\min_{\mathbf{w}} \mathcal{L}_T(\boldsymbol{\lambda}, \mathbf{w}). \quad (1)$$

The optimal model parameters $\mathbf{w}^*$ may vary with $\boldsymbol{\lambda}$, making $\mathcal{L}_V$ an *implicit* function of $\boldsymbol{\lambda}$ alone.

We approach the outer optimisation (1a) similarly to Lorraine et al. (2020) and Majumder et al. (2019), using the *hypergradient*: the total derivative of $\mathcal{L}_V$ with respect to the hyperparameters $\boldsymbol{\lambda}$. One strategy for solving (1) is therefore to alternate between updating $\mathbf{w}$ for several steps using $\frac{\partial \mathcal{L}_T}{\partial \mathbf{w}}$ and updating $\boldsymbol{\lambda}$ using the hypergradient, as shown in Figure 1 (by → and →).

Carefully distinguishing the total differential $\mathrm{d}\boldsymbol{\lambda}$ and the partial differential $\partial\boldsymbol{\lambda}$, we have

$$\frac{\mathrm{d}\mathcal{L}_V}{\mathrm{d}\boldsymbol{\lambda}} = \frac{\partial \mathcal{L}_V}{\partial \boldsymbol{\lambda}} + \frac{\partial \mathcal{L}_V}{\partial \mathbf{w}^*} \frac{\partial \mathbf{w}^*}{\partial \boldsymbol{\lambda}} . \quad (2)$$

While derivatives of $\mathcal{L}_V$ are easily computed for typical loss functions, the final derivative of the optimal model parameters ($\frac{\partial \mathbf{w}^*}{\partial \boldsymbol{\lambda}}$) presents some difficulty. Letting square brackets indicate the evaluation of the interior at the subscripted values, we may rewrite $\frac{\partial \mathbf{w}^*}{\partial \boldsymbol{\lambda}}$ as follows:

**Theorem 1 (Cauchy's Implicit Function Theorem (IFT))** *Suppose for some $\boldsymbol{\lambda}'$ and $\mathbf{w}'$ that $\left[\frac{\partial \mathcal{L}_T}{\partial \mathbf{w}}\right]_{\boldsymbol{\lambda}',\mathbf{w}'} = \mathbf{0}$. If $\frac{\partial \mathcal{L}_T}{\partial \mathbf{w}}$ is a continuously differentiable function with invertible Jacobian, then there exists a function $\mathbf{w}^*(\boldsymbol{\lambda})$ over an open subset of hyperparameter space, such that $\boldsymbol{\lambda}'$ lies in the open subset, $\left[\frac{\partial \mathcal{L}_T}{\partial \mathbf{w}}\right]_{\boldsymbol{\lambda},\mathbf{w}^*(\boldsymbol{\lambda})} = \mathbf{0}$ and*

$$\frac{\partial \mathbf{w}^*}{\partial \boldsymbol{\lambda}} = -\left(\frac{\partial^2 \mathcal{L}_T}{\partial \mathbf{w} \partial \mathbf{w}^\mathsf{T}}\right)^{-1} \frac{\partial^2 \mathcal{L}_T}{\partial \mathbf{w} \partial \boldsymbol{\lambda}^\mathsf{T}} . \quad (3)$$

$\mathbf{w}^*(\boldsymbol{\lambda})$ is called the *best response* of $\mathbf{w}$ to $\boldsymbol{\lambda}$ (Figure 1). While (3) suggests a route to computing $\frac{\partial \mathbf{w}^*}{\partial \boldsymbol{\lambda}}$, inverting a potentially high-dimensional Hessian in $\mathbf{w}$ is not computationally tractable.

## 2.2 APPROXIMATE BEST-RESPONSE DERIVATIVE

To develop and justify a computationally tractable approximation to (3), we mirror the strategy of Lorraine et al. (2020). Consider the broad class of weight optimisers with updates of the form

$$\mathbf{w}_i(\boldsymbol{\lambda}) = \mathbf{w}_{i-1}(\boldsymbol{\lambda}) - \mathbf{u}(\boldsymbol{\lambda}, \mathbf{w}_{i-1}(\boldsymbol{\lambda})) \quad (4)$$

for some arbitrary differentiable function $\mathbf{u}$, with $i$ indexing each update iteration. We deviate here from the approach of Lorraine et al. (2020) by admitting general functions $\mathbf{u}(\boldsymbol{\lambda}, \mathbf{w})$, rather than assuming the particular choice $\mathbf{u}_{\text{SGD}} = \eta \frac{\partial \mathcal{L}_T}{\partial \mathbf{w}}$ (see Appendix A.4 for more details). In particular, this allows $\boldsymbol{\lambda}$ to include optimiser hyperparameters. Differentiating (4) and unrolling the recursion gives

$$\left[\frac{\partial \mathbf{w}_i}{\partial \boldsymbol{\lambda}}\right]_{\boldsymbol{\lambda}'} = -\sum_{0 \leq j < i} \left( \prod_{0 \leq k < j} \left[\mathbf{I} - \frac{\partial \mathbf{u}}{\partial \mathbf{w}}\right]_{\boldsymbol{\lambda}', \mathbf{w}_{i-1-k}} \right) \left[\frac{\partial \mathbf{u}}{\partial \boldsymbol{\lambda}}\right]_{\boldsymbol{\lambda}', \mathbf{w}_{i-1-j}}, \tag{5}$$

where $j$ indexes our steps back through time from $\mathbf{w}_{i-1}$ to $\mathbf{w}_0$, and all $\mathbf{w}$ depend on the current hyperparameters $\boldsymbol{\lambda}'$ — see Appendix C.2 for the full derivation. Now, we follow Lorraine et al. and assume $\mathbf{w}_0, \ldots, \mathbf{w}_{i-1}$ to be equal to $\mathbf{w}_i$. With this, we simplify the product in (5) to a $j$th power by evaluating all derivatives at $\mathbf{w}_i$. This result is then used to approximate $\frac{\partial \mathbf{w}^*}{\partial \boldsymbol{\lambda}}$ by further assuming that $\mathbf{w}_i \approx \mathbf{w}^*$. These two approximations lead to

$$\left[\frac{\partial \mathbf{w}_i}{\partial \boldsymbol{\lambda}}\right]_{\boldsymbol{\lambda}'} \approx -\left[\sum_{0 \leq j < i} \left(\mathbf{I} - \frac{\partial \mathbf{u}}{\partial \mathbf{w}}\right)^j \frac{\partial \mathbf{u}}{\partial \boldsymbol{\lambda}}\right]_{\boldsymbol{\lambda}', \mathbf{w}_i(\boldsymbol{\lambda}')} \approx \left[\frac{\partial \mathbf{w}^*}{\partial \boldsymbol{\lambda}}\right]_{\boldsymbol{\lambda}'}. \tag{6}$$

We reinterpret $i$ as a predefined look-back distance, trading off accuracy and computational efficiency.

The combination of these approximations implies $\mathbf{w}_i = \mathbf{w}^*$ for all $i$, which is initially inaccurate, but which we would expect to become more correct as training proceeds. In mitigation, we perform several weight updates prior to each hyperparameter update. This means derivatives in earlier terms of the series of (6) (which are likely the largest, dominant terms) are evaluated at weights closer to $\mathbf{w}^*$, therefore making the summation more accurate. In Section 4, we show that the approximations described here result in an algorithm that is both practical and effective.

Our approximate result (6) combines the general weight update of Majumder et al. (2019) with the overall approach and constant-weight assumption of Lorraine et al. (2020). The latter empirically show that an approximation similar to (6) leads to a directionally-accurate approximate hypergradient; we illustrate the approximate updates from our derivations in Figure 1 (by ➡).

## 2.3 CONVERGENCE TO BEST-RESPONSE DERIVATIVE

To justify the approximations in (6), note that the central part of that equation is a truncated Neumann series. Taking the limit $i \to \infty$, when such a limit exists, results in the closed form

$$\left[\frac{\partial \mathbf{w}^*}{\partial \boldsymbol{\lambda}}\right]_{\boldsymbol{\lambda}'} \approx -\left[\left(\frac{\partial \mathbf{u}}{\partial \mathbf{w}}\right)^{-1} \frac{\partial \mathbf{u}}{\partial \boldsymbol{\lambda}}\right]_{\boldsymbol{\lambda}', \mathbf{w}^*(\boldsymbol{\lambda}')}. \tag{7}$$

This is precisely the result of the IFT (Theorem 1) applied to $\mathbf{u}$ instead of $\frac{\partial \mathcal{L}_T}{\partial \mathbf{w}}$; that is, substituting the simple SGD update $\mathbf{u}_{\text{SGD}}(\boldsymbol{\lambda}, \mathbf{w}) = \eta \frac{\partial \mathcal{L}_T}{\partial \mathbf{w}}$ into (7) recovers (3) exactly. Thus, under certain conditions, our approximation (6) converges to the true best-response Jacobian in the limit of infinitely long look-back windows.

## 2.4 HYPERPARAMETER UPDATES

Substituting (6) into (2) yields a tractable approximation for the hypergradient $\frac{d\mathcal{L}_V}{d\boldsymbol{\lambda}}$, with which we can update hyperparameters by gradient descent. Our implementation in Algorithm 1 (Figure 4) closely parallels Lorraine et al.'s algorithm, invoking Jacobian-vector products (Pearlmutter, 1994) during gradient computation for memory efficiency via the `grad_outputs` argument, which also provides the repeated multiplication for the $j$th power in (6). Thus, we retain the $\mathcal{O}(|\mathbf{w}| + |\boldsymbol{\lambda}|)$ time and memory cost of Lorraine et al. (2020), where $|\cdot|$ denotes cardinality. The core loop to compute the summation in (6) comes from an algorithm of Liao et al. (2018). Note that Algorithm 1 approximates $\frac{d\mathcal{L}_V}{d\boldsymbol{\lambda}}$ retrospectively by considering only the last weight update rather than any future weight updates.

Unlike differentiation-through-optimisation (Domke, 2012), Algorithm 1 crucially estimates hypergradients without reference to old model parameters, thanks to the approximate-hypergradient construction of Lorraine et al. (2020) and (6). We thus do not store network weights at multiple time

steps, so gradient-based HPO becomes possible on previously-intractable large-scale problems. In essence, we develop an approximation to online hypergradient descent (Baydin et al., 2018).

Optimiser hyperparameters generally do not affect the optimal weights, suggesting their hypergradients should be zero. However, in practice, $\mathbf{w}^*$ is better reinterpreted as the *approximately* optimal weights obtained after a finite training episode. These certainly depend on the optimiser hyperparameters, which govern the convergence of $\mathbf{w}$, thus justifying our use of the bilevel framework.

We emphasise training is not reset after each hyperparameter update — we simply continue training from where we left off, using the new hyperparameters. Consequently, Algorithm 1 avoids the time cost of multiple training restarts. While our locally greedy hyperparameter updates threaten a short-horizon bias (Wu et al., 2018), we still realise practical improvements in our experiments.

### 2.5 REINTERPRETATION OF ITERATIVE OPTIMISATION

Originally, we stated the IFT (3) in terms of minima of $\mathcal{L}_T$ (zeros of $\frac{\partial \mathcal{L}_T}{\partial \mathbf{w}}$), and substituting $\mathbf{u}_{\text{SGD}} = \eta \frac{\partial \mathcal{L}_T}{\partial \mathbf{w}}$ into (7) recovers this form of (3). However, in general, (7) recovers the Theorem for zeros of $\mathbf{u}$, which are not necessarily minima of the training loss. Despite this, our development can be compared to Lorraine et al. (2020) by expressing $\mathbf{u}$ as the derivative of an augmented 'pseudo-loss' function. Consider again the simple SGD update $\mathbf{u}_{\text{SGD}}$, which provides the weight update rule $\mathbf{w}_i = \mathbf{w}_{i-1} - \eta \frac{\partial \mathcal{L}_T}{\partial \mathbf{w}}$. By trivially defining a pseudo-loss $\overline{\mathcal{L}} = \eta \mathcal{L}_T$, we may absorb $\eta$ into a loss-like derivative, yielding $\mathbf{w}_i = \mathbf{w}_{i-1} - \frac{\partial \overline{\mathcal{L}}}{\partial \mathbf{w}}$. More generally, we may write $\overline{\mathcal{L}} = \int \mathbf{u}(\boldsymbol{\lambda}, \mathbf{w}) \, \mathrm{d}\mathbf{w}$.

Expressing the update in this form suggests a reinterpretation of the role of optimiser hyperparameters. Conventionally, our visualisation of gradient descent has the learning rate control the size of steps over some undulating landscape. Instead, we propose fixing a unit step size, with the 'learning rate' scaling the landscape underneath. Similarly, we suppose a 'momentum' could, at every point, locally squash the loss surface in the negative-gradient direction and stretch it in the positive-gradient direction. In aggregate, these transformations straighten out optimisation trajectories and bring local optima closer to the current point. While more complex hyperparameters lack a clear visualisation in this framework, it nevertheless allows a broader class of hyperparameters to 'directly alter the loss function' instead of remaining completely independent, circumventing the problem with optimiser hyperparameters noted by Lorraine et al. (2020). Figure 2 and Appendix A.3 support this argument.

## 3 RELATED WORK

Kohavi & John (1995) first noted different problems respond optimally to different hyperparameters; Hutter et al. (2018, Chapter 1) summarise the resulting *hyperparameter optimisation* (HPO) field.

Black-box HPO treats the training process as atomic, selecting trial configurations by grid search (coarse or intractable at scale), random search (Bergstra & Bengio (2012); often more efficient) or population-based searches (mutating promising trials). Pure Bayesian Optimisation (Močkus et al., 1978; Snoek et al., 2012) guides the search with a predictive model; many works seek to exploit its sample efficiency (e.g. Swersky et al. (2014a); Snoek et al. (2015); Wang et al. (2016); Kandasamy et al. (2017); Lévesque et al. (2017); Perrone et al. (2019)). However, these methods require each proposed configuration to be fully trained, incurring considerable computational expense. Other techniques infer information *during* a training run — from learning curves (Provost et al., 1999; Swersky et al., 2014b; Domhan et al., 2015; Chandrashekaran & Lane, 2017; Klein et al., 2017), smaller surrogate problems (Petrak, 2000; van den Bosch, A. et al., 2004; Krueger et al., 2015; Sparks et al., 2015; Thornton et al., 2013; Sabharwal et al., 2016) or intelligent resource allocation (Jamieson & Talwalkar, 2016; Li et al., 2017; Falkner et al., 2018; Bertrand et al., 2017; Wang et al., 2018). Such techniques could be applied on top of our algorithm to improve performance.

In HPO with nested *bilevel* optimisation, hyperparameters are optimised conditioned on optimal weights, enabling updates *during* training, with a separate validation set mitigating overfitting risk; this relates to meta-learning (Franceschi et al., 2018). Innovations include differentiable unrolled stochastic gradient descent (SGD) updates (Domke, 2012; Maclaurin et al., 2015; Baydin et al., 2018; Shaban et al., 2019; Majumder et al., 2019), conjugate gradients or hypernetworks (Lorraine & Duvenaud, 2018; Lorraine et al., 2020; MacKay et al., 2019; Fu et al., 2017), solving one level while penalising suboptimality of the other (Mehra & Hamm, 2020), and deploying Cauchy's implicit

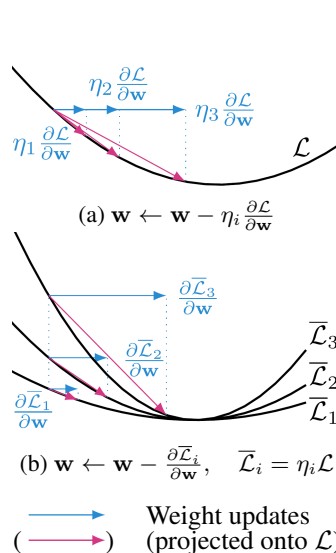

(a) $\mathbf{w} \leftarrow \mathbf{w} - \eta_i \frac{\partial \mathcal{L}}{\partial \mathbf{w}}$

(b) $\mathbf{w} \leftarrow \mathbf{w} - \frac{\partial \overline{\mathcal{L}}_i}{\partial \mathbf{w}}, \quad \overline{\mathcal{L}}_i = \eta_i \mathcal{L}$

$\longrightarrow$ Weight updates
$(\longrightarrow)$ (projected onto $\mathcal{L}$)

Figure 2: Reinterpreting the role of learning rate $\eta$ over a loss function $\mathcal{L}$. (a) In the classical setting, $\eta$ scales the gradient of some fixed $\mathcal{L}$. (b) In our setting, $\eta$ scales $\mathcal{L}$ to form a 'pseudo-loss' $\overline{\mathcal{L}}$, whose gradient is used as-is: our loss function has become dependent on $\eta$. The same weight updates are obtained in both (a) and (b).

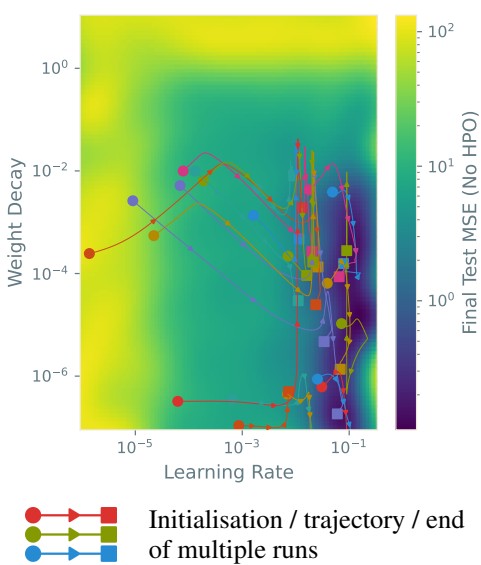

Initialisation / trajectory / end of multiple runs

Figure 3: Sample hyperparameter trajectories from training UCI Energy under $Our^{WD+LR}$ configuration. Background shading gives a non-HPO baseline with hyperparameters fixed at the corresponding initial point; these results are interpolated by a Gaussian process. Note the trajectories are generally attracted to the valley of high performance at learning rates around $10^{-1}$ and weight decays below $10^{-2}$.

function theorem (Larsen et al., 1996; Bengio, 2000; Luketina et al., 2016; Lorraine et al., 2020). Donini et al. (2020) extrapolate training to optimise arbitrary learning rate schedules, extending earlier work on gradient computation (Franceschi et al., 2017), but do not immediately accommodate other hyperparameters. While non-smooth procedures exist (Lopez-Ramos & Beferull-Lozano, 2021), most methods focus on smooth hyperparameters which augment the loss function (e.g. weight regularisation) and work with the augmented loss directly. Consequently, they cannot handle optimiser hyperparameters not represented in the loss function (e.g. learning rates and momentum factors; Lorraine et al. (2020)). Many of these methods compute hyperparameter updates locally (not over the entire training process), which may induce short-horizon bias (Wu et al., 2018), causing myopic convergence to local optima.

Modern developments include theoretical reformulations of bilevel optimisation to improve performance (Liu et al., 2020; Li et al., 2020a), optimising distinct hyperparameters for each model parameter (Lorraine et al., 2020; Jie et al., 2020), and computing forward-mode hypergradient averages using more exact techniques than we do (Micaelli & Storkey, 2020). Although these approaches increase computational efficiency and the range of tunable parameters, achieving both benefits at once remains challenging. Our algorithm accommodates a diverse range of differentiable hyperparameters, but retains the efficiency of existing approaches (specifically Lorraine et al. (2020)). Inevitably, we also inherit gradient-based methods' difficulties with robustness and discrete hyperparameters.

## 4 EXPERIMENTS

Our empirical evaluation uses the hardware and software detailed in Appendix A.1, with code available at `https://github.com/rmclarke/OptimisingWeightUpdateHyperparameters`.

Throughout, we train models using SGD with weight decay and momentum. We uniformly sample initial learning rates, weight decays and momenta, using logarithmic and sigmoidal transforms (see Appendix B.3), applying each initialisation in the following eight settings:

**Algorithm 1**
Scalable One-Pass Optimisation of High-Dimensional Weight-Update Hyperparameters by Implicit Differentiation

**while** training continues **do**
  **for** $t \leftarrow 1$ **to** $T$ **do**     ▷ *T steps of weight updates*
    $\mathbf{w} \leftarrow \mathbf{w} - \mathbf{u}(\boldsymbol{\lambda}, \mathbf{w})$
  **end for**
  $\mathbf{p} = \mathbf{v} = \left[\frac{\partial \mathcal{L}_V}{\partial \mathbf{w}}\right]_{\boldsymbol{\lambda}, \mathbf{w}}$     ▷ *Initialise accumulators*
  **for** $j \leftarrow 1$ **to** $i$ **do**    ▷ *Accumulate first summand in* (6)
    $\mathbf{v} \leftarrow \mathbf{v} - \texttt{grad}(\mathbf{u}(\boldsymbol{\lambda}, \mathbf{w}), \mathbf{w}, \texttt{grad\_outputs} = \mathbf{v})$
    $\mathbf{p} \leftarrow \mathbf{p} + \mathbf{v}$
  **end for**     ▷ *Now* $\mathbf{p} \approx \left[\frac{\partial \mathcal{L}_V}{\partial \mathbf{w}} \left(\frac{\partial \mathbf{u}}{\partial \mathbf{w}}\right)^{-1}\right]_{\boldsymbol{\lambda}, \mathbf{w}}$
  $\mathbf{g}_{\text{indirect}} = -\texttt{grad}(\mathbf{u}(\boldsymbol{\lambda}, \mathbf{w}), \boldsymbol{\lambda}, \texttt{grad\_outputs} = \mathbf{p})$
     ▷ *Now* $\mathbf{g}_{\text{indirect}} \approx -\left[\frac{\partial \mathcal{L}_V}{\partial \mathbf{w}} \left(\frac{\partial \mathbf{u}}{\partial \mathbf{w}}\right)^{-1} \frac{\partial \mathbf{u}}{\partial \boldsymbol{\lambda}}\right]_{\boldsymbol{\lambda}, \mathbf{w}}$
  $\boldsymbol{\lambda} \leftarrow \boldsymbol{\lambda} - \kappa \left(\left[\frac{\partial \mathcal{L}_V}{\partial \boldsymbol{\lambda}}\right]_{\boldsymbol{\lambda}, \mathbf{w}} + \mathbf{g}_{\text{indirect}}\right)$
     ▷ *Any gradient-based optimiser (SGD shown, LR $\kappa$)*
**end while**

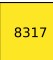

(a) Baseline: *Random* ($\times 1\,000$)

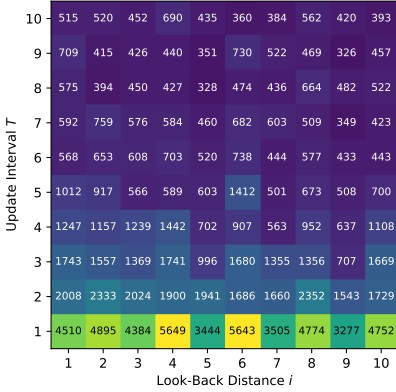

(b) Our approach: $Ours^{WD+LR+M}$ ($\times 1\,000$)

Figure 4: Left: Pseudocode for our method. Right: Median final test loss on UCI Energy after 400 hyperparameter updates from random initialisations, for various update intervals $T$ and look-back distances $i$, with (a) no hyper-parameter tuning, *Random*, and (b) our proposed method, $Ours^{WD+LR+M}$.

**Random** No HPO; hyperparameters held constant at their initial values, as in random search.
**Random ($\times$ LR)** *Random* with an extra hyperparameter increasing or decreasing the learning rate by multiplication after every $T = 10$ update steps (emulating a hyperparameter update).
**Random (3-batched)** *Random* reprocessed to retain only the best result of every three runs. Allows *Random* to exceed our methods' computational budgets; imitates population training.
**Lorraine** Lorraine et al. (2020)'s method optimising weight decay; other hyperparameters constant.
**Baydin** Baydin et al. (2018)'s method optimising learning rate only; other hyperparameters constant.
**Ours$^{WD+LR}$** Algorithm 1 updating weight decay and learning rate; other hyperparameters constant.
**Ours$^{WD+LR+M}$** Optimising all hyperparameters (adding momentum to $Ours^{WD+LR}$)
**Ours$^{WD+HDLR+M}$** $Ours^{WD+LR+M}$ with independent learning rates for each model parameter
**Diff-through-Opt** Optimising all hyperparameters using exact hypergradients (Domke, 2012)

Any unoptimised hyperparameters are fixed at their random initial values. Ideally, we seek resilience to poor initialisations, which realistically arise in unguided hyperparameter selection. Hyperparameters are tuned on the validation set; this is combined with the training set for our *Random* settings, so each algorithm observes the same data. We use the UCI/Kin8nm dataset split sizes of Gal & Ghahramani (2016) and standard 60%/20%/20% splits for training/validation/test datasets elsewhere, updating hyperparameters every $T = 10$ batches with look-back distance $i = 5$ steps (except *Baydin*, which has no such *meta-hyperparameters*, so updates hyperparameters at every batch (Baydin et al., 2018)).

Numerical data is normalised to zero-mean, unit-variance. For efficiency, the computational graph is detached after each hyperparameter update, so we never differentiate through hyperparameter updates — essentially, back-propagation and momentum histories are truncated. Our approximate hypergradient is passed to Adam (Kingma & Ba, 2015) with meta-learning rate $\kappa = 0.05$ and default $\beta_1 = 0.9, \beta_2 = 0.999$. While these meta-hyperparameters are not tuned, previous work indicates performance is progressively less sensitive to higher-order hyperparameters (Franceschi et al., 2017; 2018; Majumder et al., 2019). As some settings habitually chose unstable high learning rates, we clip these to $[10^{-10}, 1]$ throughout.

**UCI Energy: Proof of Concept** First, we broadly illustrate Algorithm 1 on UCI Energy, using a one-layer multi-layer perceptron (MLP) with 50 hidden units and ReLU (Glorot et al., 2011) activation functions, trained under $Ours^{WD+LR}$ for $4\,000$ full-batch epochs (see Appendix B.1 for details). Figure 3 shows the evolution of learning rate and weight decay from a variety of initialisations, overlaid on the performance obtained when all hyperparameters are kept fixed during training. Notice the trajectories are attracted towards the region of lowest test loss, indicating our algorithm is capable of useful learning rate and weight decay adjustment.

Table 1: Final test MSEs after training UCI and Kin8nm datasets for 4 000 full-batch epochs from each of 200 random hyperparameter initialisations, showing best and bootstrapped average performance. Uncertainties are standard errors; bold values lie in the error bars of the best algorithm.

| Method | UCI Energy | | | Kin8nm ($\times 1\,000$) | | | UCI Power | | |
|---|---|---|---|---|---|---|---|---|---|
| | Mean | Median | Best | Mean | Median | Best | Mean | Median | Best |
| Random | 24 ±2 | 8.3 ±0.7 | 0.124 | 36 ±2 | 30 ±2 | **5.70** | 70 ±7 | 20 ±1 | **15.4** |
| Random ($\times$ LR) | 35 ±3 | 13 ±3 | 0.113 | 44 ±2 | 41 ±3 | 6.18 | 106 ±8 | 29 ±5 | 16.0 |
| Random (3-batched) | 5 ±1 | 2 ±1 | **0.104** | 15 ±1 | 10 ±1 | 6.24 | 17.9 ±0.4 | 17.0 ±0.1 | 15.5 |
| Lorraine | 24 ±2 | 8.7 ±0.9 | 0.145 | 36 ±2 | 30 ±3 | 6.32 | 69 ±6 | 19.7 ±0.9 | 15.6 |
| Baydin | 5.5 ±0.2 | 6.8 ±0.1 | 0.144 | 23.4 ±0.8 | 26 ±1 | 6.00 | 17.80 ±0.07 | 17.94 ±0.07 | 15.4 |
| Ours$^{WD+LR}$ | 2.1 ±0.1 | 1.8 ±0.2 | 0.130 | **8.1** ±0.2 | 7.60 ±0.07 | 6.21 | 17.26 ±0.02 | 17.21 ±0.01 | 15.6 |
| Ours$^{WD+LR+M}$ | 0.96 ±0.08 | 0.30 ±0.03 | 0.112 | **8.1** ±0.4 | 7.10 ±0.08 | 5.90 | 17.19 ±0.01 | 17.192 ±0.004 | 15.6 |
| Ours$^{WD+HDLR+M}$ | **0.6** ±0.2 | **0.28** ±0.01 | 0.139 | **8.1** ±0.3 | 7.47 ±0.06 | 6.04 | **16.69** ±0.03 | **16.76** ±0.02 | 15.4 |
| Diff-through-Opt | 0.93 ±0.08 | 0.34 ±0.04 | 0.132 | **7.9** ±0.3 | **7.13** ±0.07 | 5.88 | 17.15 ±0.02 | 17.184 ±0.006 | 15.5 |

## 4.1 UCI and Kin8nm Datasets: Establishing Robustness

Next, we consider the same 50-hidden-unit MLP applied to seven standard UCI datasets (Boston Housing, Concrete, Energy, Naval, Power, Wine, Yacht) and Kin8nm, in a fashion analogous to Gal & Ghahramani (2016) (we do not consider dropout or Bayesian formulations). We train 200 hyperparameter initialisations for 4 000 full-batch epochs. Table 1 shows results for three datasets, with complete loss evolution and distribution plots in Appendix B.4. Some extreme hyperparameters caused numerical instability and NaN final losses, which our averages ignore. NaN results are not problematic: they indicate extremely poor initialisations, which should be easier for the user to rectify than merely mediocre hyperparameters. Error bars are based on 1 000 sets of bootstrap samples.

Given the sparse distribution of strong hyperparameter initialisations, random sampling unsurprisingly achieves generally poor averages. *Random ($\times$ LR)* applies a learning rate multiplier, uniformly chosen in $[0.95, 1.01]$; these limits allow extreme initial learning rates to revert to more typical values after 400 multiplications. This setting's poor performance shows naïve learning rate schedules cannot match our algorithms' average improvements. A stronger baseline is *Random (3-batched)*, which harshly simulates the greater computational cost of our methods by retaining only the best of every three *Random* trials (according to validation loss). This setting comes close to, but cannot robustly beat, our methods — a claim reinforced by final test loss distributions (Figure 9, Appendix B.4).

Lorraine et al. (2020)'s algorithm is surprisingly indistinguishable from *Random* in our trials, though it must account for three random hyperparameters while varying only one (the weight decay). We surmise that learning rates and momenta are more important hyperparameters to select, and poor choices cannot be overcome by intelligent use of weight decay. Baydin et al. (2018)'s algorithm, however, comes closer to the performance of *Random (3-batched)*, indicating more successful intervention in its sole optimisable hyperparameter (learning rate). Variance is also much lower than the preceding algorithms, suggesting greater stability. That said, despite concentrating on a more important hyperparameter, *Baydin* still suffers from being unable to control every hyperparameter.

Our scalar algorithms (*Ours$^{WD+LR}$* and *Ours$^{WD+LR+M}$*) appear generally more robust to these initialisations, with average losses beating *Random*, *Lorraine* and *Baydin*. Figures 8 and 9 (Appendix B.4) clearly distinguish these algorithms from the preceding: we achieve performant results over a wider space of initial hyperparameters. Given it considers more hyperparameters, *Ours$^{WD+LR+M}$* predictably outperforms *Ours$^{WD+LR}$*, although the difference is less prominent in certain datasets. Unlike pure HPO, the non-*Random* algorithms in Table 1 vary hyperparameters *during training*, combining aspects of HPO and schedule learning. They are thus more flexible than conventional, static-hyperparameter techniques, even in high dimensions (Lorraine et al., 2020).

*Diff-through-Opt* exactly differentiates the current loss with respect to the hyperparameters, over the same $i = 5$ look-back window and $T = 10$ update interval. As an exact version of *Ours$^{WD+LR+M}$* (though subject to the same short-horizon bias), it unsurprisingly matches the other algorithms (Figures 8 and 9, Appendix B.4). However, our scalar methods' proximity to this exact baseline is reassuring given our much-reduced memory requirements and generally comparable error bars. In these experiments, lengthening *Diff-through-Opt*'s look-back horizon to all 4 000 training steps, and repeating those steps for 30 hyperparameter updates, did not improve its performance (Appendix B.7).

$Ours^{WD+HDLR+M}$ theoretically mitigates short-horizon bias (Wu et al., 2018) by adapting appropriately to high- and low-curvature directions. While this substantially improves performance on some datasets, it is outperformed by scalar methods on others (Figures 8 and 9, Appendix B.4). We also see slightly reduced stability, with more trials diverging to NaN final losses. Intuitively, the risk of overfitting to the validation set is expected to increase as we introduce more hyperparameters, which could explain this behaviour — committing to large learning rates in a few selected directions, as this method often does, may prove harmful if the loss surface dynamics change suddenly. However, we leave detailed investigation of the high-dimensional dynamics to future work.

## 4.2 LARGE-SCALE DATASETS: PRACTICAL SCALABILITY

**Fashion-MNIST: HPO in Multi-Layer Perceptrons**   We train the same single 50-unit hidden layer MLP on 10 epochs of Fashion-MNIST (Xiao et al., 2017), using 50-sample batches. Table 2 and Figure 12a (Appendix B.6) show average test set cross-entropies over 100 initialisations. Clearly-outlying final losses (above $10^3$) are set to NaN to stop them dominating our error bars.

Echoing Section 4.1, for arbitrary hyperparameter initialisations, our methods generally converge more robustly to lower losses, even when (as in Figure 12a, Appendix B.6) NaN solutions are included in our statistics. Importantly, we see mini-batches provide sufficient gradient information for our HPO task. *Diff-through-Opt*'s failure to beat our methods is surprising; we suppose noisy approximate gradients may regularise our algorithms, preventing them from seeking the short-horizon optimum so directly, thus mitigating short-horizon bias (see our sensitivity study in Section 4.3). *Diff-through-Opt* with long look-back horizons does not improve performance for equal computation (Appendix B.7). Finally, median and best loss evolution plots are shown in Figures 5b and 5c, the latter including results for a Bayesian Optimisation baseline. For more details, see Appendices B.5 and B.6.

**Penn Treebank: HPO in Recurrent Networks**   Now, we draw inspiration from Lorraine et al. (2020)'s large-scale trials: a 2-layer, 650-unit LSTM (Hochreiter & Schmidhuber, 1997) with learnable embedding, trained on the standard Penn Treebank-3-subset benchmark dataset (Marcus et al., 1999) for 72 epochs. To focus our study, we omit the dropout, activation regularisation and predefined learning rate schedules used by Lorraine et al., though we retain training gradient clipping to a Euclidean norm of $0.25$. Training considers length-70 subsequences of 40 parallel sequences, using 50 random hyperparameter initialisations.

Table 2 and Figure 12b (Appendix B.6) show final test perplexities. They reflect our intuition that adjusting progressively more hyperparameters reduces average test losses, continuing the trend we have seen thus far. Highly bimodal final loss distributions for some algorithms cause wide bootstrap-sampled error bars. Learning rate adjustments show particular gains: *Our* algorithms and *Diff-through-Opt* perform particularly well in less-optimal configurations.

**CIFAR-10: HPO in Convolutional Networks**   Finally, to demonstrate scalability, we train a ResNet-18 (He et al., 2016) on CIFAR-10 (Krizhevsky, 2009) for 72 epochs. We use the unaugmented dataset (since unbiased data augmentation over both training and validation datasets exhausts our GPU memory) and optimise hyperparameters as before, using 100-image batches.

Table 2 and Figure 12c (Appendix B.6) show our results. Our gains are now more marginal, with this setting apparently robust to its initialisation, though the general ranking of algorithms remains similar, and we retain useful improvements over our baselines. However, our best final accuracies fall short of state-of-the-art, suggesting a more intricate and clever setting may yield further performance gains.

Table 2: Final test *cross-entropy ($^{\dagger}$perplexity) on larger datasets. Bold values are the lowest in class.

| Method | Fashion-MNIST* | | | Penn Treebank$^{\dagger}$ | | | CIFAR-10* | | |
|---|---|---|---|---|---|---|---|---|---|
| | Mean | Median | Best | Mean | Median | Best | Mean | Median | Best |
| Random | 0.92 ±0.07 | 0.60 ±0.08 | 0.340 | 4700 ±700 | 3000 ±3000 | 169 | 1.90 ±0.05 | 1.9 ±0.1 | 0.819 |
| Random (× LR) | 1.29 ±0.08 | 1.2 ±0.3 | 0.339 | 5700 ±600 | 7000 ±2000 | 200 | 1.77 ±0.05 | 1.70 ±0.09 | 0.847 |
| Random (3-batched) | 0.8 ±0.1 | 0.49 ±0.09 | 0.349 | 1300 ±600 | 470 ±80 | 171 | 1.60 ±0.07 | 1.55 ±0.09 | 0.817 |
| Lorraine | 0.95 ±0.07 | 0.61 ±0.08 | 0.343 | 4700 ±600 | 3000 ±3000 | 170 | 1.41 ±0.07 | 1.26 ±0.07 | **0.688** |
| Baydin | 0.409 ±0.004 | 0.413 ±0.002 | 0.337 | 4200 ±600 | 2000 ±2000 | 149 | 1.60 ±0.05 | 1.6 ±0.1 | 0.756 |
| Ours$^{WD+LR}$ | 0.373 ±0.002 | 0.375 ±0.003 | **0.336** | 360 ±20 | 360 ±30 | 139 | 1.17 ±0.02 | 1.170 ±0.005 | 0.715 |
| Ours$^{WD+LR+M}$ | **0.369** ±0.002 | **0.369** ±0.001 | 0.339 | **270** ±30 | **210** ±50 | **100** | **1.13** ±0.02 | **1.16** ±0.01 | 0.712 |
| Ours$^{WD+HDLR+M}$ | 0.398 ±0.004 | 0.386 ±0.003 | 0.350 | 310 ±20 | 270 ±10 | 150 | 1.33 ±0.05 | 1.32 ±0.07 | 0.822 |
| Diff-through-Opt | 0.386 ±0.001 | 0.385 ±0.002 | 0.355 | **300** ±10 | 290 ±20 | 114 | 1.195 ±0.009 | 1.197 ±0.007 | 0.967 |

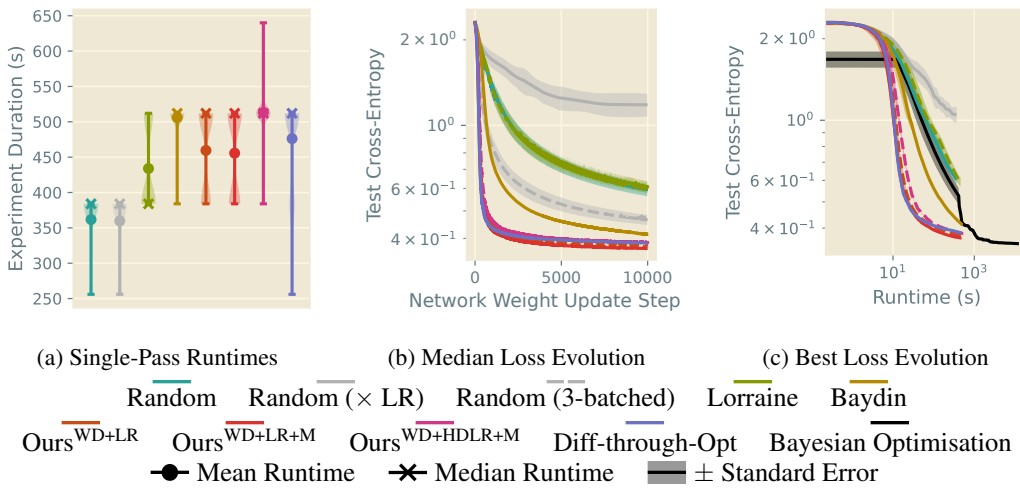

(a) Single-Pass Runtimes      (b) Median Loss Evolution      (c) Best Loss Evolution

Random   Random ($\times$ LR)   Random (3-batched)   Lorraine   Baydin

Ours$^{WD+LR}$   Ours$^{WD+LR+M}$   Ours$^{WD+HDLR+M}$   Diff-through-Opt   Bayesian Optimisation

$\bullet$ Mean Runtime   $\times$ Median Runtime   $\pm$ Standard Error

Figure 5: Illustrations of training a one-layer 50-unit MLP on Fashion-MNIST over 100 random hyperparameter initialisations. We include a *Bayesian Optimisation* baseline from Appendix B.5 and loss evolution plots from Appendix B.6.

### 4.3 MISCELLANEOUS STUDIES

**Experiment Runtimes**   For our larger-scale experiments, we illustrate comparative runtimes in Figure 14 (Appendix B.6.1, where they are discussed in detail). Note from our Fashion-MNIST sample (Figure 5a) that all HPO algorithms have a lower computational cost than naïvely training two typical fixed hyperparameter initialisations, despite achieving substantial HPO effect. This compares extremely favourably to HPO methods relying on repeated retraining. Additional experiments in Appendix B.8.2 combine *Ours$^{WD+LR+M}$* with Asynchronous Hyperband (Li et al., 2020b) or Population-Based Training (Jaderberg et al., 2017), showing improved and unchanged performance/time trade-offs, respectively.

**UCI Energy: Sensitivity Study**   Finally, we consider a range of update intervals $T$ and look-back distances $i$ on UCI Energy, performing 400 hyperparameter updates from 100 random initialisations on each, using *Ours$^{WD+LR+M}$*. We plot the median final test losses for each choice of $T$ and $i$ to the right of Figure 4, and also the median performance with no HPO (*Random*), which we outperform in every case. Performance generally improves with larger $T$ — likely because the total number of weight updates increases with $T$, since the number of hyperparameter updates is fixed at 400. While increasing $i$ gives more terms in our approximating series (6), these extra terms become more and more biased: they are evaluated at the current weight values instead of at the progressively more different past weight values. We theorise this trade-off explains the more complex relationship between $i$ and final performance. More details are given in Appendix B.2.

## 5 CONCLUSION AND FUTURE WORK

We have presented an algorithm for optimising continuous hyperparameters, specifically those appearing in a differentiable weight update, including optimiser hyperparameters. Our method requires only a single training pass, has a motivating true-hypergradient convergence argument, and demonstrates practical benefits at a range of experimental scales without greatly sacrificing training time. We also tackle traditionally-intractable per-parameter learning rate optimisation on non-trivial datasets. However, this setting surprisingly underperformed its scalar counterpart; further work is necessary to understand this result.

As future work, our myopic approach could be extended to longer horizons by incorporating the principles of recent work by Micaelli & Storkey (2020), which presents a promising research direction. We also depend inconveniently on meta-hyperparameters, which are not substantially tuned. Ultimately, we desire systems which are completely independent of human configuration, thus motivating investigations into the removal of these settings.

## SOCIETAL IMPACT / ETHICS STATEMENT

Our work fits into the broad subfield of automatic machine learning (AutoML), which aims to use automation to strip away the tedious work that is necessary to implement practical ML systems. Our method focuses on automating the oft-labelled 'black art' of optimising hyperparameters. This contributes towards the democratisation of ML techniques, which we hope will improve accessibility to non-experts.

However, our method also has some associated risks. For one, developers of unethical machine learning applications (e.g. for mass surveillance, identity theft or automated weaponry) may use our techniques to improve their systems' performance. The heavily metric-dominated nature of our field creates additional concerns — for instance, end-users of our method may not appreciate that optimising naïvely for training and validation loss alone may result in dangerously poorly-trained or unethical models if the chosen metric, developer's intentions and moral good do not align.

More broadly, users may rely excessively on HPO techniques to optimise their models' performance, which can lead to poor results and inaccurate comparisons if the HPO strategy is imperfect. Further, reducing the need to consider hyperparmeter tuning abstracts away an important component of how machine learning methods work in practice. Knowledge of this component may then become less accessible, inhibiting understanding and future research insights from the wider community.

Our datasets are drawn from standard benchmarks, and thus should not introduce new societal risks. Similarly, our work aims to decrease the computational burden of HPO, which should mitigate the environmental impact of ML training — an ever more important goal in our increasingly environmentally-conscious society.

## REPRODUCIBILITY STATEMENT

All datasets we use are publicly available; for the Penn Treebank dataset, we provide a link in Table 4 (Appendix A.2). What little data processing we perform is fully explained in the corresponding subsection of Section 4.

Our mathematical arguments are presented fully and with disclosure of all assumptions in Section C.

Source code for all our experiments is provided to reviewers, and is made available on GitHub (`https://github.com/rmclarke/OptimisingWeightUpdateHyperparameters`). This source code contains a complete description of our experimental environment, configuration files and instructions on the reproduction of our experiments.

## ACKNOWLEDGEMENTS

We acknowledge computation provided by the Cambridge Service for Data Driven Discovery (CSD3) operated by the University of Cambridge Research Computing Service (`www.csd3.cam.ac.uk`), provided by Dell EMC and Intel using Tier-2 funding from the Engineering and Physical Sciences Research Council (capital grant EP/P020259/1), and DiRAC funding from the Science and Technology Facilities Council (`www.dirac.ac.uk`).

We thank Austin Tripp for his feedback on drafts of this paper.

Ross Clarke acknowledges funding from the Engineering and Physical Sciences Research Council (project reference 2107369, grant EP/S515334/1).

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

# A    NOTES

## A.1    EXPERIMENTAL CONFIGURATION

Table 3: System configurations used to run our experiments.

| Type | CPU | GPU (NVIDIA) | Python | PyTorch | CUDA |
|---|---|---|---|---|---|
| Consumer Desktop | Intel Core i7-3930K | RTX 2080GTX | 3.9.7 | 1.8.1 | 10.1 |
| Local Cluster | Intel Core i9-10900X | RTX 2080GTX | 3.7.12 | 1.8.1 | 10.1 |
| Cambridge Service for Data Driven Discovery (CSD3)* | AMD EPYC 7763 | Ampere A100 | 3.7.7 | 1.10.1 | 11.1 |

\* `www.csd3.cam.ac.uk`

Our experiments are variously performed on one of three sets of hardware, as detailed in Table 3. While system configurations varied between experiments, this does not impair comparability. All Penn Treebank and CIFAR-10 experiments were performed on the CSD3 Cluster; otherwise, all experiments comparing runtimes were performed on the Local Cluster. We make use of GPU acceleration throughout, with the PyTorch (Paszke et al., 2019) and Higher (Grefenstette et al., 2019) libraries.

On CIFAR-10 only, a minor bug in our data processing code meant the mean and standard deviation used to normalise the dataset were incorrect by a factor of 255. As batch normalisation in the ResNet-18 model largely mitigates this error, and every algorithm was provided with the same mal-normalised data, this does not threaten the validity of our results. We retain the incorrect transformation in our published code for reproducibility, but have clearly annotated the affected lines.

## A.2    DATASET LICENCES

Table 4: Licences under which we use datasets in this work.

| Dataset | Licence | Source |
|---|---|---|
| UCI Boston | | |
| UCI Concrete | | |
| UCI Energy | | |
| Kin8nm | Unknown; widely used | Gal & Ghahramani (2016, Github) |
| UCI Naval | | |
| UCI Power | | |
| UCI Wine | | |
| UCI Yacht | | |
| Fashion-MNIST | MIT | PyTorch via `torchvision` |
| Penn Treebank | Proprietary; fair use subset, widely used | * |
| CIFAR-10 | No licence specified | PyTorch via `torchvision` |

\* `http://www.fit.vutbr.cz/~imikolov/rnnlm/simple-examples.tgz`

Our datasets are all standard in the ML literature. For completeness, we outline the licences under which they are used in Table 4.

## A.3    EXPANDED INTUITIVE REINTERPRETATION OF ITERATIVE OPTIMISATION

In this section, we take the opportunity to give a more verbose elaboration on the ideas presented in Section 2.5, which seeks to address methodological concerns raised by Lorraine et al. (2020) and reinterpret iterative optimisation in light of our derived algorithm.

Lorraine et al. (2020) base their derivations on the training loss $\mathcal{L}_T$, which does not depend on optimiser hyperparameters such as the learning rate or momentum coefficient. Differentiating $\mathcal{L}_T$ with respect to these hyperparameters thus cannot provide any useful information for computing their gradient-based updates. Similarly, the Implicit Function Theorem cannot be applied to motivate their approximation in this case, since $\mathcal{L}_T$ is not an implicit function of these hyperparameters. Consequently, Lorraine et al. declare their method is methodologically restricted to operate on hyperparameters on which the training loss depends directly.

In contrast, we work with an arbitrary update $\mathbf{u}$, which *does* then depend on the optimiser hyperparameters, allowing us to take the corresponding derivatives and apply the Implicit Function Theorem. However, the optimisation objective remains the same, so we should rightly be suspicious that the methodological issue raised by Lorraine et al. may still apply. To address these remaining concerns, we look to recast our development in the form of Lorraine et al. (2020). We do this by defining a 'pseudo-loss' $\overline{\mathcal{L}}$ such that our weight updates in (4) are equivalent to

$$\mathbf{w} \leftarrow \mathbf{w} - \left[\frac{\partial \overline{\mathcal{L}}}{\partial \mathbf{w}}\right]_{\boldsymbol{\lambda}, \mathbf{w}}. \tag{8}$$

We must acknowledge that our optimisation objective is now for the update $\mathbf{u}$ to be zero (equivalent to a stationary point of $\overline{\mathcal{L}}$), rather than the training loss $\mathcal{L}_T$ being at a minimum. But with that caveat, we may apply the machinery of Lorraine et al. directly: we have constructed a 'loss function' which depends directly on all the hyperparameters, so may legitimately deploy their strategy. By comparison with (4), we have that, for any particular problem,

$$\overline{\mathcal{L}} = \int \mathbf{u}(\boldsymbol{\lambda}, \mathbf{w}) \, \mathrm{d}\mathbf{w}. \tag{9}$$

Often, in simple cases, the mathematical construction in (9) has a geometric interpretation. Consider the elementary case of vanilla SGD with learning rate $\eta$, which gives the update rule $\mathbf{u}_{\text{SGD}} = \eta \frac{\partial \mathcal{L}_T}{\partial \mathbf{w}}$. The required pseudo-loss is then $\overline{\mathcal{L}} = \eta \mathcal{L}_T$, which suggests two equivalent geometric interpretations of the optimisation problem we are solving.

In the first (conventional) interpretation, we have a fixed loss surface $\mathcal{L}_T$ whose minimum we seek. To optimise $\eta$ in this problem, we imagine standing on the loss surface at our current point, then consider the effect of taking different-sized steps across it. Clearly, this interpretation decouples $\eta$ from the underlying function $\mathcal{L}_T$, so suffers from Lorraine et al.'s methodological issue.

In the second (our novel) interpretation, we seek the minimum of the loss surface $\overline{\mathcal{L}}$, which we may traverse only by taking steps of precisely the magnitude of the local gradient. Now, when we optimise $\eta$, we are *rescaling* the surface beneath our update step, and consequently also rescaling the update step size. In this way, we have coupled $\eta$ and $\mathcal{L}_T$ into one objective $\overline{\mathcal{L}}$, which varies with $\eta$ and so adheres to the methodological restrictions noted by Lorraine et al.. The key idea is thus that we have reinterpreted the problem to allow optimiser hyperparameters to dynamically transform the objective surface, which legitimises our derivation.

When we consider SGD with momentum, the situation becomes more complicated, because there is not an immediately obvoius closed form for $\overline{\mathcal{L}}$. Suppose we are descending a steep-sided valley of the original loss surface $\mathcal{L}_T$. The momentum acts to 'pull' the update step in the direction it has accumulated in its buffer (down the axis of the valley), so it acts to augment gradients in the valley's downhill direction, and diminishes the effect of gradients in other directions.

If we are working with the pseudo-loss $\overline{\mathcal{L}}$, we need to achieve the same effect, but we do not have direct control over the update step size, which is determined by the local gradient of $\overline{\mathcal{L}}$. So if the update step is aligned with the accumulated momentum, we must 'squash' the loss surface about our current point in this direction, such that our fixed update step lands at a point further down the valley in the original, unscaled space of $\mathcal{L}_T$. Accordingly, if the update step is *mis*aligned with the accumulated momentum, this squashing in the direction of the momentum buffer must be combined with a 'stretching' in the update step direction, so that we make *less* progress in that direction of the original space. In aggregate, this is a complex and intricate transformation of the space, such that we bring the 'correct' new point (as specified by the optimiser) to lie underneath the update step given by $-\frac{\partial \overline{\mathcal{L}}}{\partial \mathbf{w}}$.

To summarise, the key principle we seek to convey in Section 2.5 is that of creating a new pseudo-loss surface $\overline{\mathcal{L}}$, which is a transformed version of $\mathcal{L}_T$, such that the update steps taken by a hyperparameter-free optimiser over the pseudo-loss $\overline{\mathcal{L}}$ are equivalent to those taken by the true optimiser over the original training loss $\mathcal{L}_T$.

### A.4 EXPANDED DISCUSSION OF UPDATE FUNCTION $\mathbf{u}(\boldsymbol{\lambda}, \mathbf{w})$

Throughout our derivation, we refer to an arbitrary weight update function $\mathbf{u}(\boldsymbol{\lambda}, \mathbf{w})$. While we have not expressed this in its most general form for clarity, we outline here that this formulation supports a wide array of non-trivial weight update optimisation algorithms beyond SGD.

$\mathbf{u}(\boldsymbol{\lambda}, \mathbf{w})$ is precisely the quantity which is added to the network weights $\mathbf{w}$ during an update step. For instance, in the PyTorch framework (Paszke et al., 2019), the pre-implemented optimisers first compute some update tensor $\boldsymbol{\Delta}\mathbf{w}$, then perform the final operation $\mathbf{w} \leftarrow \mathbf{w} - \boldsymbol{\Delta}\mathbf{w}$ to update the weights. Thus, we have $\mathbf{u}(\boldsymbol{\lambda}, \mathbf{w}) = \boldsymbol{\Delta}\mathbf{w}$, and automatic differentiation trivially gives the necessary derivatives to apply our algorithm.

Some optimisers have internal state, for instance the momentum buffer of SGD with momentum and the two rolling average buffers of Adam. Although this dependency is not notated in our presentation of $\mathbf{u}$, it does not invalidate our development — the only extra care required is to detach these buffers from the computational graph after each hyperparameter update. For PyTorch's implementation of SGD with learning rate $\eta$, momentum coefficient $\mu$ and weight decay coefficient $\xi$, we thus have

$$\mathbf{v} \leftarrow \mu\mathbf{v} + \frac{\partial \mathcal{L}_T}{\partial \mathbf{w}} + \xi\mathbf{w} \qquad\qquad \mathbf{u}_{\text{momentum}} = \eta\mathbf{v}\,, \tag{10}$$

and need simply take care to detach $\mathbf{v}$ from the computational graph after each update. For Adam, we may read the definition of $\mathbf{u}$ straight from its original paper (Kingma & Ba, 2015); in their notation:

$$\mathbf{u}_{\text{Adam}} = \alpha \cdot \frac{\hat{m}_t}{\sqrt{\hat{v}_t} + \epsilon}\,, \tag{11}$$

where $t$ indexes time and $\hat{m}_t, \hat{v}_t$ are internal states, and we simply detach those states from the computational graph periodically. Those states depend directly on the additional states $m_t, v_t$, and ultimately on the gradient $\nabla_\theta f_t(\theta_{t-1})$ (which is $\left[\frac{\partial \mathcal{L}_T}{\partial \mathbf{w}}\right]_{\mathbf{w}_{i-1}}$ in our notation).

# B ADDITIONAL RESULTS

## B.1 UCI ENERGY: PROOF OF CONCEPT

Our proof of concept graphic (Figure 3) is constructed using the same experimental configuration as Section 4.1. Training 500 random initialisations of the model, weight decay and learning rate for 4 000 network weight update steps on *Random* (without any HPO) yields source data for the heatmap forming the background of Figure 3. The heatmap itself is the predictive mean of a Gaussian process (Williams & Rasmussen, 1996) using the sum of an RBF and white noise kernel. We fit the Gaussian process in log (input and output) space using the defaults of the Scikit-learn implementation (Pedregosa et al., 2011), which automatically select hyperparameters for the kernels. Any NaN data points are replaced with a large final test loss (150) for the purpose of computing this regression. We then train a small number of additional initialisations using $Ours^{WD+LR}$ and plot these trajectories. As always, the training set provides network weight updates, the validation set provides hyperparameter updates and our results plot final performance on the test set. While we do not expect the optimal trajectory during training to coincide with gradients of our background heatmap, the latter provides useful context.

Generally, given our limited training budget, larger learning rates improve final performance up to a point, then cause instability if taken to extremes, and the movement of our trajectories reflects this. Many trajectories select large weight decay coefficients before doubling back on themselves to smaller weight decays — suggestive of our algorithm using the weight decay to avoid converging to specific minima. Trajectories generally converge to the valley of good test performance at a learning rate of around $10^{-1}$ and weight decay below $10^{-2}$, further demonstrating appropriate behaviour. In short, we are able to make sensible and interpretable updates to the hyperparameters of this problem, which lends support to the applicability of our method.

## B.2 UCI ENERGY: HYPERGRADIENT STUDIES

Our sensitivity studies are based on a similar experimental configuration to Section 4.1: for each choice of update interval $T$ and look-back distance $i$, we generate 100 random initialisations of weight decay, learning rate and momentum. Each initialisation is trained for 400 hyperparameter steps, so increasing $T$ permits more network weight updates; averaging over the 100 final test losses yields each result cell in Figure 6 and Figure 7. As in Section 4.1, we consider the $Ours^{WD+LR+M}$ and *Diff-through-Opt* algorithms, optimising hyperparameters using Adam with hyper-learning rate $\kappa = 0.05$ and truncating gradient histories at the previous hyperparameter update.

B.2.1 SENSITIVITY OF UPDATE INTERVAL AND LOOK-BACK DISTANCE

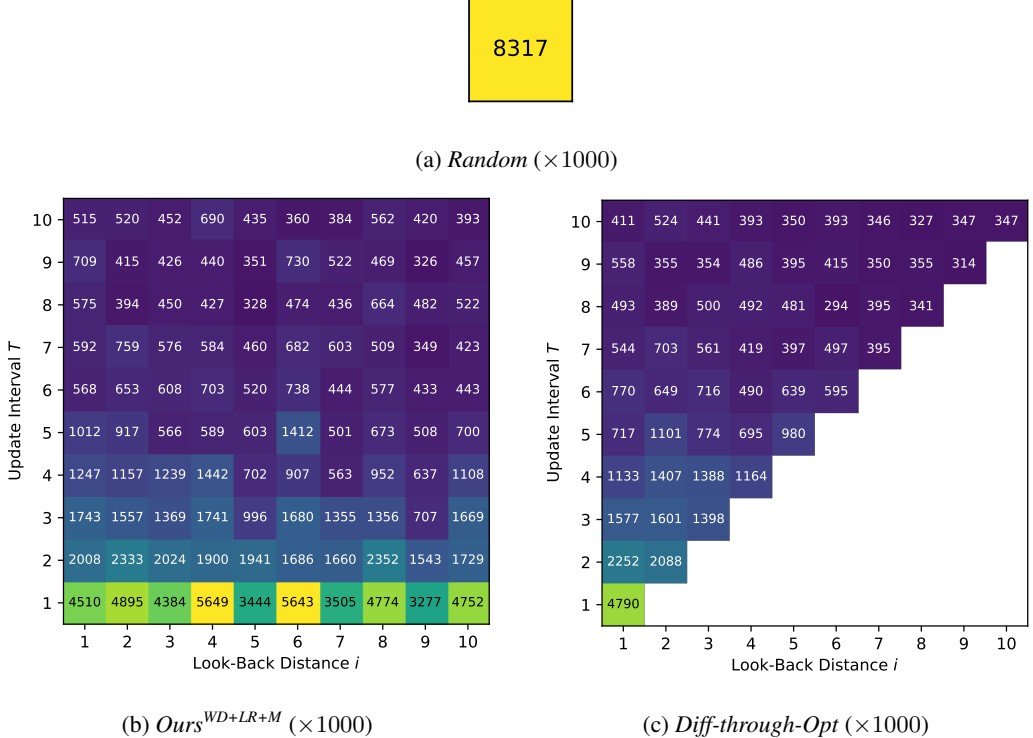

(a) *Random* ($\times 1000$)

(b) *Ours$^{WD+LR+M}$* ($\times 1000$)

(c) *Diff-through-Opt* ($\times 1000$)

Figure 6: Median final test loss over 100 repetitions, after optimising UCI Energy for 400 hyperparameter update steps, from a variety of update intervals $T$ and look-back distances $i$.

Figure 6 shows our final performance results in this setting: our algorithm's strongest configurations (Figure 6b) compare favourably with results obtained using the exact hypergradients of *Diff-through-Opt* (Figure 6c). Noting that the conditions necessary to guarantee our algorithm's convergence to the best-response derivative (specifically that the current network weights are optimal for the current hyperparameters) are almost certainly not met in practice, it is unsurprising that *Diff-through-Opt* achieves greater performance. This difference is much smaller, however, than that between both HPO algorithms and the *Random* base case; given our substantial efficiency benefits over *Diff-through-Opt*, we make an acceptable compromise between accuracy and scalability. Further, these results justify our choice of $T = 10$ and $i = 5$ to match the setting of Lorraine et al. (2020). That said, recall these figures enforce consistency of number of hyperparameter updates, not of total computational expense; there remains a case for using large numbers of less precise hyperparameter updates (small $T$ and $i$), which may promote faster convergence in practice.

Our results illustrate the trade-off in selecting a look-back distance $i$. In principle, increasing $i$ includes more terms in our Neumann series approximation to the best-response derivative (6), which should result in closer convergence to the true derivative. But our derivation of this expression replaces the sequence $\mathbf{w}_0, \mathbf{w}_1, \cdots, \mathbf{w}_{i-1}$ from (5) with $\mathbf{w}_i$ at all time steps. Because the $j$th summation step of (5) involves a product of terms in $\mathbf{w}_{i-1}$ back to $\mathbf{w}_{i-1-j}$, the approximation of each summation step in (6) becomes less and less accurate as $j$ increases, since we expect earlier model weights to become more different from $\mathbf{w}_i$. A larger $i$ causes us to continue the summation up to larger $j$, so introduces progressively less accurate terms into our approximations. Thus, increasing $i$ is only of value while this inaccuracy is counteracted by the improved convergence of our summation, giving the behaviour we see here.

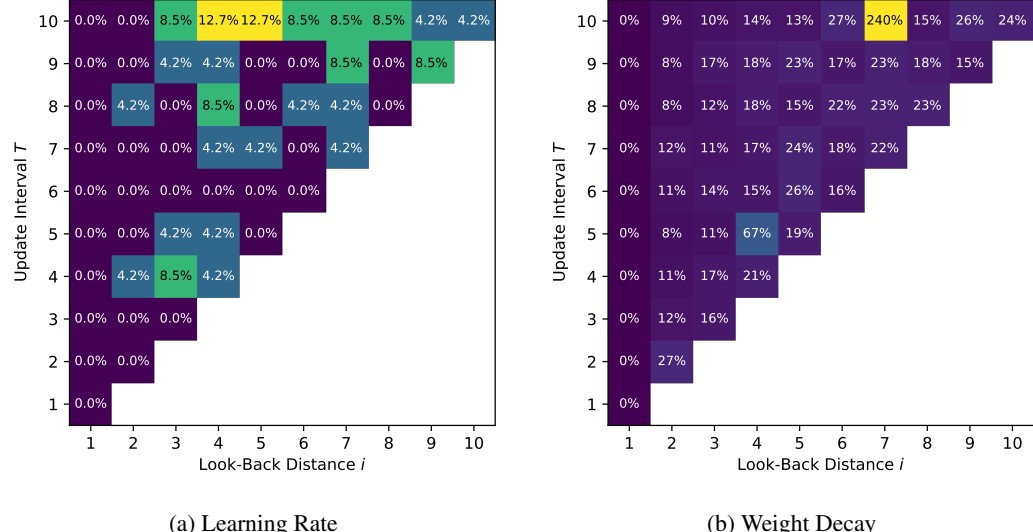

(a) Learning Rate    (b) Weight Decay

Figure 7: Mean absolute error in our approximate hypergradients, based on the first hyperparameter updates of 50 repetitions for each cell, using UCI Energy. We vary the update interval $T$ and look-back distance $i$ to investigate the sensitivity of the problem to these meta-hyperparameters. Hypergradients are shown separately for each hyperparameter.

### B.2.2 ACCURACY OF HYPERGRADIENTS

We use this same experimental setting to evaluate the accuracy of our approximate hypergradients, but disable learning rate clipping so as to allow the natural behaviour of each algorithm to be displayed. . Our approximate and exact settings are only guaranteed to be trying to compute the same hypergradient at the very first hyperparameter update, so we consider this time step only. Using those first hyperparameter updates from 50 random initialisations of the hyperparameters and model, we compute the absolute error of the approximate hypergradient ($Ours^{WD+LR+M}$) as a percentage of the exact hypergradient (*Diff-through-Opt*), then plot the means of these errors in Figure 7.

Notice our learning rate hypergradients do not exhibit any particularly strong trends over the meta-hyperparameters we consider, but our approximate weight decay hypergradients seem to benefit from a shorter look-back distance. Some of the latter average proportional errors are large; a closer examination of the error distributions reveals these cells have outlying results at high errors. We hypothesise that the generally very small weight decay coefficients present more opportunity for numerical computational error than the generally larger learning rates, in which case hypergradients for different hyperparameters may have markedly different accuracies. Indeed, the apparently quantised errors in the learning rate hypergradients indicate precision truncation is influencing the results of this study. However, our previous coarser studies suggest that, despite being approximate, our method does not suffer catastrophically from inaccurate hypergradients.

## B.3 EXPERIMENT DETAILS

Table 5: Transformation and initialisation parameters used in our hyperparameter optimisation.

| Hyperparameter | Initialisation Space | Initialisation Range | | Optimisation Space | Natural Range | |
|---|---|---|---|---|---|---|
| | | Min | Max | | Min | Max |
| Learning Rate | Logarithmic | −6 | −1 | Logarithmic | $1 \times 10^{-6}$ | 0.1 |
| Weight Decay | Logarithmic | −7 | −2 | Logarithmic | $1 \times 10^{-7}$ | 0.01 |
| Momentum | Natural | 0 | 1 | Sigmoidal | 0 | 1 |

Since the hyperparameters associated with SGD only take meaningful values on subsets of the real line, we apply gradient descent to transformed versions of the hyperparameters. For non-negative hyperparameters, such as learning rate and weight decay, we work with the base-10 logarithm of the actual value; for momentum, which lies in the unit interval $[0, 1]$, we work with the inverse-sigmoid of the actual value. These transformed search spaces mean a naïve implementation of gradient descent will not update hyperparameters to nonsensical values.

Our hyperparameter initialisations are drawn uniformly from a broad range, intended to be a superset of typical values to permit analysis of the realistic case where we have little intuition for the optimal choices. Thus, our hyperparameter ranges include typical values for a variety of applications, while giving each algorithm the opportunity to briefly stretch into more extreme values should these be useful on occasion. Where we have applied a logarithmic transformation to a hyperparameter (learning rate and weight decay), we draw samples uniformly in logarithmic space; where inverse-sigmoid transforms are used (momentum), we draw samples in natural space to ensure adequate exploration of values near 0 and 1. Table 5 summarises our initialisation and optimisation strategy.

Throughout, we seek robustness to a range of different initialisations, which is indicated by average final performance across a variety of initial hyperparameter choices. We include metrics of best performance to show each algorithm is capable of reaching the same vicinity as the best possible case, but this is not the focus of our work. Intuitively, a more naïve algorithm (e.g. *Random*) could stumble across the best possible final model weights, in which case no other algorithm would be able to beat it; this would not necessarily mean we would prefer the *Random* algorithm in a general setting.

## B.4 UCI AND KIN8NM DATASETS

Results from all UCI datasets we considered and Kin8nm are presented in Figures 8 and 9. These show many of the features already discussed in Section 4.1. We account for experimental runs ending in NaN losses by appropriately scaling the maximum value of the CDFs in Figure 9 to be less than 1.0. For each dataset, one random configuration is selected, and the resulting learning rate evolutions plotted in Figure 10. Note that UCI Naval generated a large number of NaN runs, which explains the prominent failure of certain algorithms to reach the top of our CDF plots.

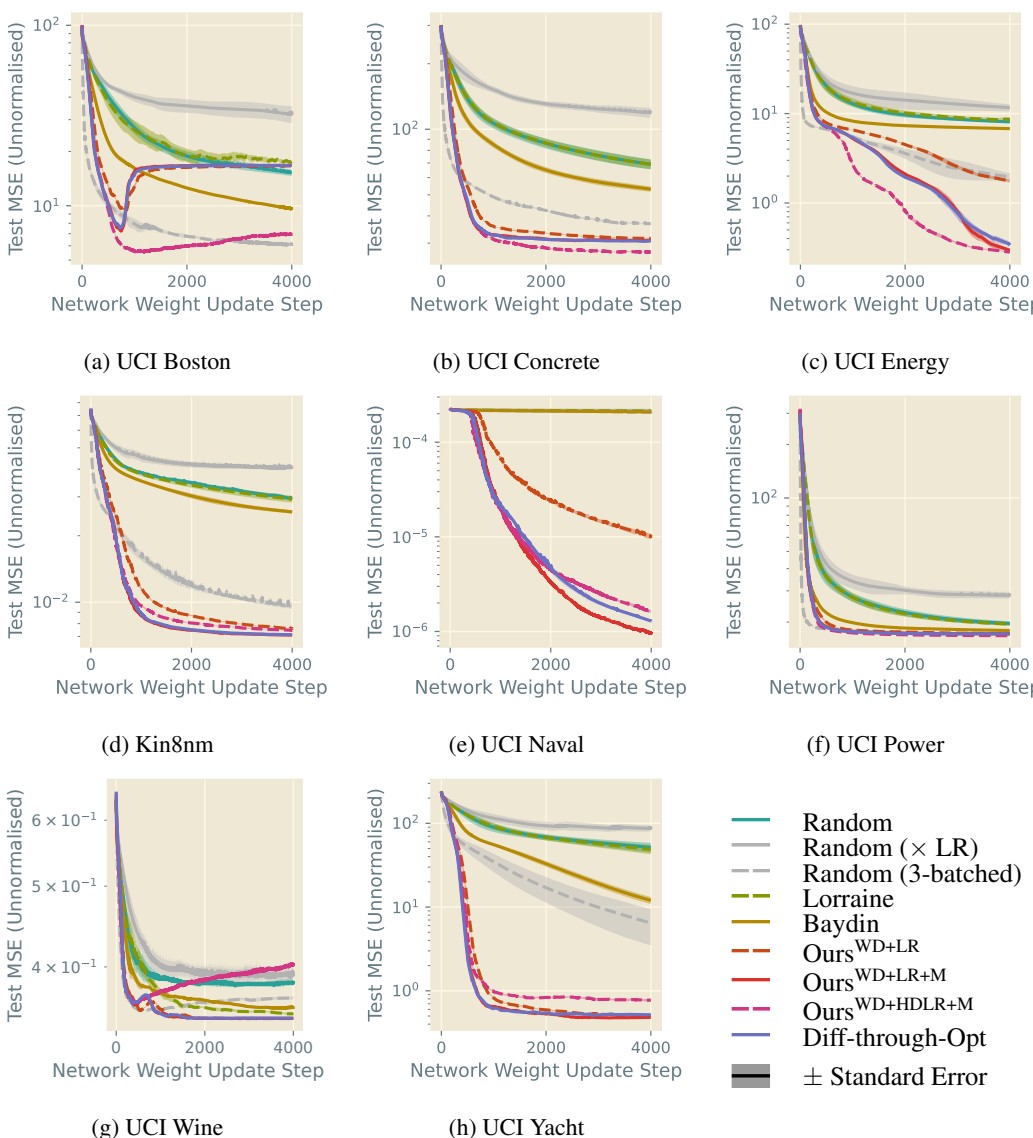

Figure 8: Evolution of test MSEs during training of UCI and Kin8nm datasets for 4 000 full-batch epochs from each of 200 random initialisations of SGD learning rate, weight decay and momentum. We bootstrap sample our results to construct a median test MSE estimator at each update step and its standard error.

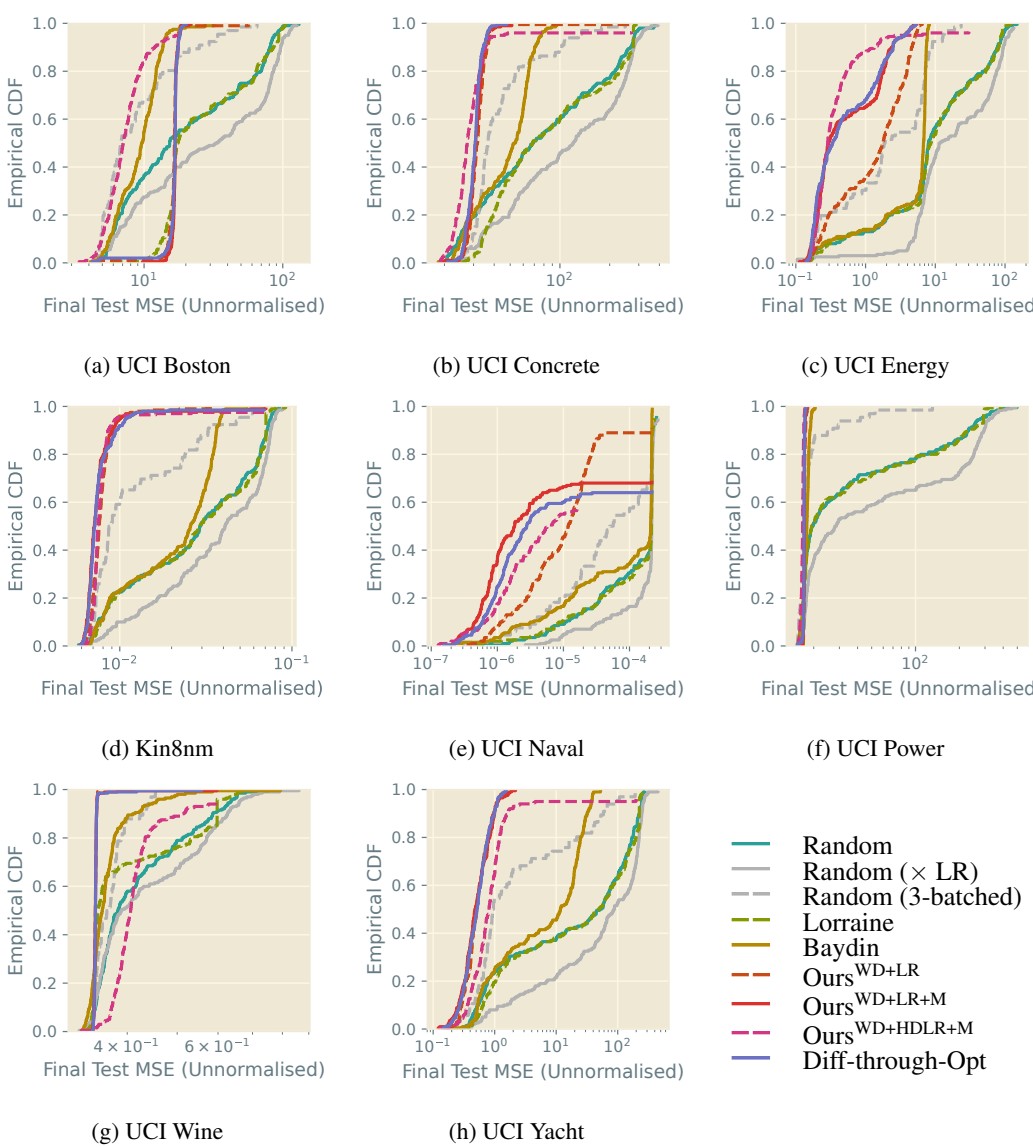

Figure 9: Empirical CDFs of final test losses, after training UCI and Kin8nm datasets for 4 000 full-batch epochs from each of 200 random initialisations of SGD learning rate, weight decay and momentum. Runs ending with NaN losses account for the CDFs not reaching 1.0. Higher curves are better; note the logarithmic horizontal scales.

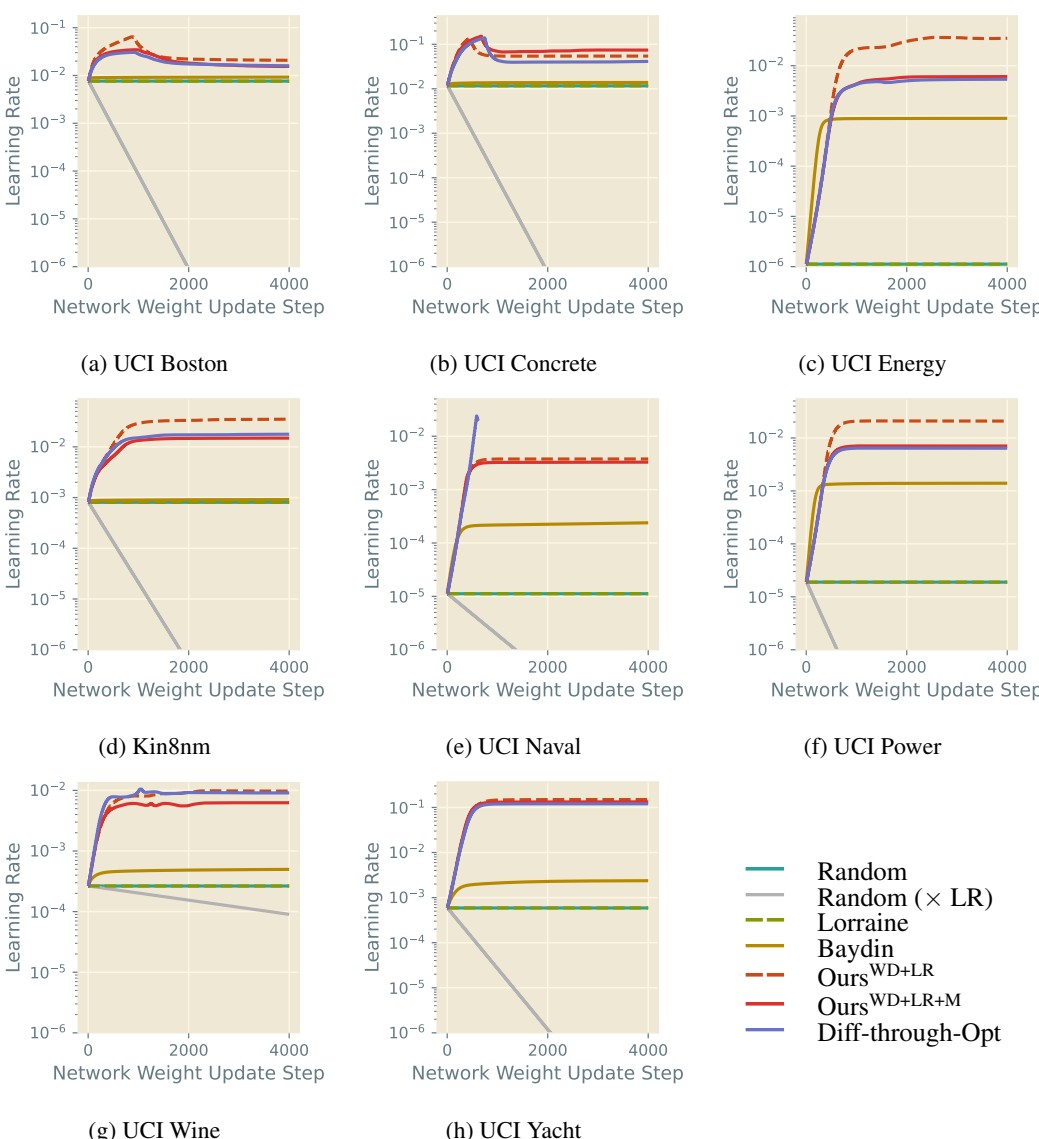

Figure 10: Sample learning rate evolutions after training UCI and Kin8nm datasets for 4 000 full-batch epochs; one of 200 random initialisations is shown for each dataset, chosen randomly from our results.

## B.5 UCI, KIN8NM AND FASHION-MNIST: BAYESIAN OPTIMISATION BASELINES

Table 6: Repeat of Table 1: final test MSE on selected UCI and Kin8nm datasets. Now includes 32 repetitions of 30-sample Bayesian Optimisation, which is not considered in our emboldening.

| Method | UCI Energy | | | Kin8nm ($\times 1\,000$) | | | UCI Power | | |
|---|---|---|---|---|---|---|---|---|---|
| | Mean | Median | Best | Mean | Median | Best | Mean | Median | Best |
| Random | 24 $\pm 2$ | 8.3 $\pm 0.7$ | 0.124 | 36 $\pm 2$ | 30 $\pm 2$ | **5.70** | 70 $\pm 7$ | 20 $\pm 1$ | **15.4** |
| Random ($\times$ LR) | 35 $\pm 3$ | 13 $\pm 3$ | 0.113 | 44 $\pm 2$ | 41 $\pm 3$ | 6.18 | 106 $\pm 8$ | 29 $\pm 5$ | 16.0 |
| Random (3-batched) | 5 $\pm 1$ | 2 $\pm 1$ | **0.104** | 15 $\pm 1$ | 10 $\pm 1$ | 6.24 | 17.9 $\pm 0.4$ | 17.0 $\pm 0.1$ | 15.5 |
| Lorraine | 24 $\pm 2$ | 8.7 $\pm 0.9$ | 0.145 | 36 $\pm 2$ | 30 $\pm 3$ | 6.32 | 69 $\pm 6$ | 19.7 $\pm 0.9$ | 15.6 |
| Baydin | 5.5 $\pm 0.2$ | 6.8 $\pm 0.1$ | 0.144 | 23.4 $\pm 0.8$ | 26 $\pm 1$ | 6.00 | 17.80 $\pm 0.07$ | 17.94 $\pm 0.07$ | 15.4 |
| Ours$^{WD+LR}$ | 2.1 $\pm 0.1$ | 1.8 $\pm 0.2$ | 0.130 | **8.1** $\pm 0.2$ | 7.60 $\pm 0.07$ | 6.21 | 17.26 $\pm 0.02$ | 17.21 $\pm 0.01$ | 15.6 |
| Ours$^{WD+LR+M}$ | 0.96 $\pm 0.08$ | 0.30 $\pm 0.03$ | 0.112 | **8.1** $\pm 0.4$ | **7.10** $\pm 0.08$ | 5.90 | 17.19 $\pm 0.01$ | 17.192 $\pm 0.004$ | 15.6 |
| Ours$^{WD+HDLR+M}$ | **0.6** $\pm 0.2$ | **0.28** $\pm 0.01$ | 0.139 | **8.1** $\pm 0.3$ | 7.47 $\pm 0.06$ | 6.04 | **16.69** $\pm 0.03$ | **16.76** $\pm 0.02$ | 15.4 |
| Diff-through-Opt | 0.93 $\pm 0.08$ | 0.34 $\pm 0.04$ | 0.132 | **7.9** $\pm 0.3$ | **7.13** $\pm 0.07$ | 5.88 | 17.15 $\pm 0.02$ | 17.184 $\pm 0.006$ | 15.5 |
| Bayesian Optimisation | 0.4 $\pm 0.1$ | 0.23 $\pm 0.03$ | 0.106 | 7.3 $\pm 0.2$ | 6.9 $\pm 0.2$ | 6.22 | 16.08 $\pm 0.09$ | 16.2 $\pm 0.2$ | 15.0 |

Table 7: Partial repeat of Table 2: final test cross-entropy on Fashion-MNIST. Now includes 32 repetitions of 30-sample Bayesian Optimisation, which is not considered in our emboldening.

| Method | Fashion-MNIST | | |
|---|---|---|---|
| | Mean | Median | Best |
| Random | 0.92 $\pm 0.07$ | 0.60 $\pm 0.08$ | 0.340 |
| Random ($\times$ LR) | 1.29 $\pm 0.08$ | 1.2 $\pm 0.3$ | 0.339 |
| Random (3-batched) | 0.8 $\pm 0.1$ | 0.49 $\pm 0.09$ | 0.349 |
| Lorraine | 0.95 $\pm 0.07$ | 0.61 $\pm 0.08$ | 0.343 |
| Baydin | 0.409 $\pm 0.004$ | 0.413 $\pm 0.002$ | 0.337 |
| Ours$^{WD+LR}$ | 0.373 $\pm 0.002$ | 0.375 $\pm 0.003$ | **0.336** |
| Ours$^{WD+LR+M}$ | **0.369** $\pm 0.002$ | **0.369** $\pm 0.001$ | 0.339 |
| Ours$^{WD+HDLR+M}$ | 0.398 $\pm 0.004$ | 0.386 $\pm 0.003$ | 0.350 |
| Diff-through-Opt | 0.386 $\pm 0.001$ | 0.385 $\pm 0.002$ | 0.355 |
| Bayesian Optimisation | 0.3501 $\pm 0.0005$ | 0.3505 $\pm 0.0007$ | 0.344 |

To contextualise our methods against the best possible performance, we also perform 32 repetitions of 30-sample Bayesian Optimisation (BO) on selected UCI datasets, Kin8nm and Fashion-MNIST, using the same procedure as *Random* to train each sampled configuration (but now with an separate validation set used to compare configurations), and record these in Tables 6 and 7. We use the Python BO implementation of Nogueira (2014), retaining the default configuration that the first five of our 30 initialisations be drawn randomly from the space. All runtimes are measured based on 8-way parallel execution per GPU. Although BO involves a much greater computational burden than the other methods we consider, it should more robustly estimate the optimal hyperparameters.

Our average performance falls short of the BO benchmark, but we come substantially closer than the *Random* and *Lorraine* baselines, and similarly as close as *Diff-through-Opt*. On Fashion-MNIST especially, our methods are very competitive, achieving performance in the vicinity of BO with 30 times less computation. Considering our best runs, we seem to suffer only a minor performance penalty (compared to the best hyperparameters we found) by adopting Algorithm 1.

As an additional comparison, we show in Figure 11 the evolution of incumbent best test loss over wall time during our previous training of 200 (UCI and Kin8nm)/100 (Fashion-MNIST) initialisations. Clearly we would expect BO's greater computation to result in smaller final losses, which we observe, but our methods are capable of reaching performant results much faster than BO. This underscores the principal benefit of our algorithm: with only a single training episode and a relatively small runtime penalty, we achieve a meaningful HPO effect, which reaches the vicinity of the ideal performance found by much slower and more computationally onerous methods.

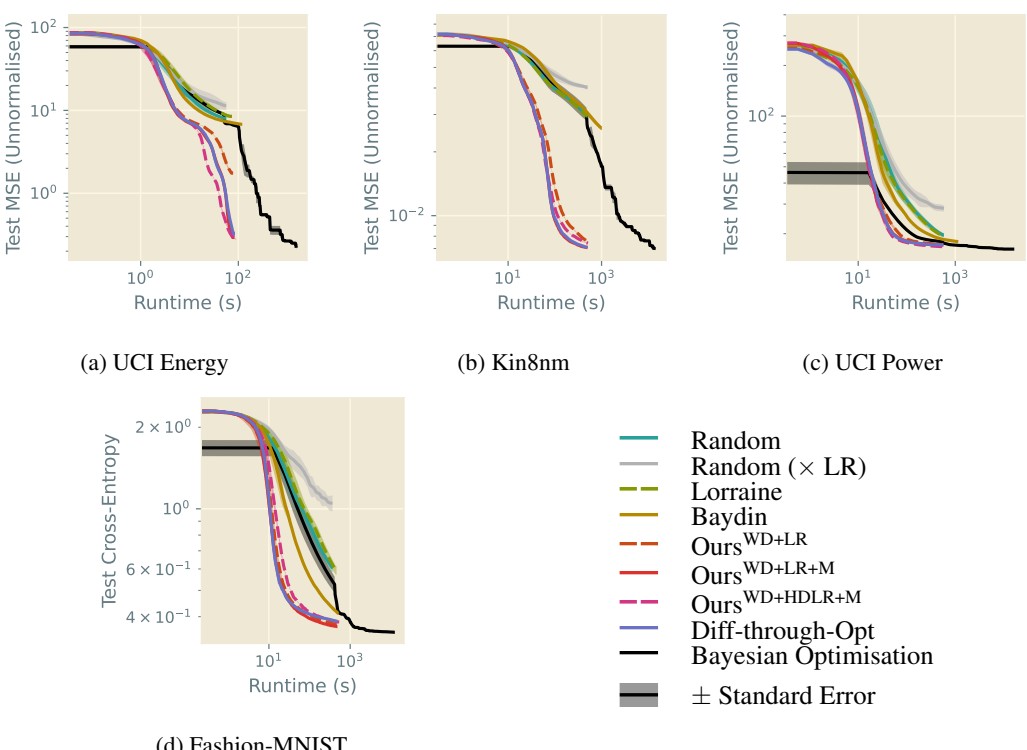

Figure 11: Evolution of incumbent best test MSE with time, during training of selected UCI datasets/Kin8nm/Fashion-MNIST from 200/200/100 random hyperparameter initialisations, with Bayesian Optimisation for comparison. Shaded envelopes are bootstrapped standard errors of the best loss found at that time. Note the logarithmic horizontal axes.

### B.6  LARGE-SCALE DATASETS

For Fashion-MNIST, Penn Treebank and CIFAR-10, Figure 12 shows the empirical distributions of final test performance, contextualising the values shown in Table 2. In addition, Figure 13 and Table 8 show our results in terms of error percentages for Fashion-MNIST and CIFAR-10.

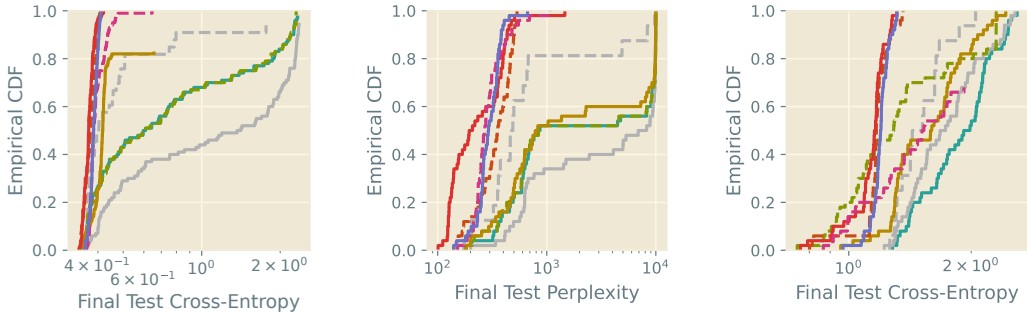

(a) Fashion-MNIST using an MLP     (b) Penn Treebank using an LSTM     (c) CIFAR-10 using a ResNet-18

Figure 12: Empirical CDFs of final test losses, trained on larger-scale datasets for $50^{(a)}/72^{(b, c)}$ epochs from $100^{(a)}/50^{(b, c)}$ random SGD hyperparameter initialisations. Key as in Figure 13; notes as in Figure 9.

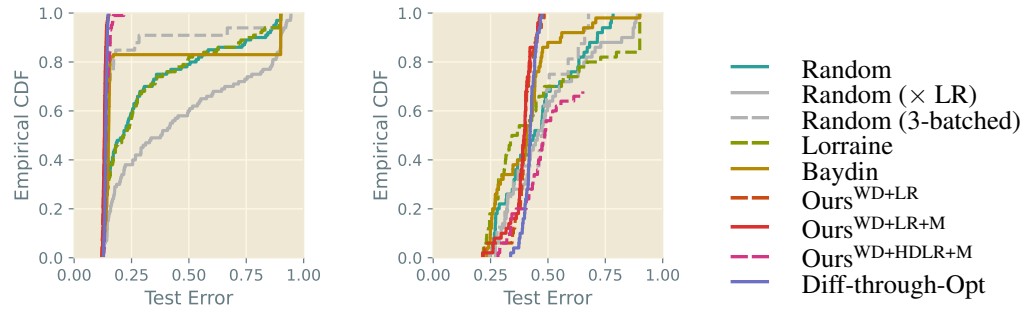

(a) Fashion-MNIST using an MLP     (b) CIFAR-10 using a ResNet-18

Figure 13: Empirical CDFs of final test errors after training on larger-scale datasets for $50^{(a)}/72^{(b)}$ epochs from each of $100^{(a)}/50^{(b)}$ random initialisations of SGD hyperparameters. Notes as in Figure 9.

Table 8: Quantitative analysis of final test errors from Figure 13. Bold values are the lowest in class.

| Method | Fashion-MNIST | | | CIFAR-10 | | |
|---|---|---|---|---|---|---|
| | Mean | Median | Best | Mean | Median | Best |
| Random | 0.31 ± 0.02 | 0.21 ± 0.02 | 0.116 | 0.46 ± 0.02 | 0.44 ± 0.03 | 0.243 |
| Random (× LR) | 0.46 ± 0.03 | 0.37 ± 0.06 | **0.115** | 0.50 ± 0.03 | 0.46 ± 0.03 | 0.244 |
| Random (3-batched) | 0.20 ± 0.03 | 0.141 ± 0.009 | 0.119 | 0.41 ± 0.03 | **0.39** ± 0.07 | 0.245 |
| Lorraine | 0.32 ± 0.02 | 0.22 ± 0.03 | 0.120 | 0.46 ± 0.03 | **0.36** ± 0.04 | 0.206 |
| Baydin | 0.27 ± 0.03 | 0.1499 ± 0.0008 | 0.117 | 0.40 ± 0.02 | 0.41 ± 0.02 | 0.220 |
| Ours$^{WD+LR}$ | 0.1329 ± 0.0007 | 0.133 ± 0.001 | 0.118 | **0.393** ± 0.008 | 0.399 ± 0.004 | 0.207 |
| Ours$^{WD+LR+M}$ | **0.1318** ± 0.0006 | **0.1316** ± 0.0006 | 0.119 | **0.386** ± 0.008 | 0.396 ± 0.005 | **0.204** |
| Ours$^{WD+HDLR+M}$ | 0.143 ± 0.001 | 0.1384 ± 0.0008 | 0.122 | 0.44 ± 0.02 | 0.45 ± 0.02 | 0.268 |
| Diff-through-Opt | 0.1376 ± 0.0005 | 0.1372 ± 0.0008 | 0.125 | 0.420 ± 0.004 | 0.422 ± 0.004 | 0.331 |

### B.6.1 RUNTIMES

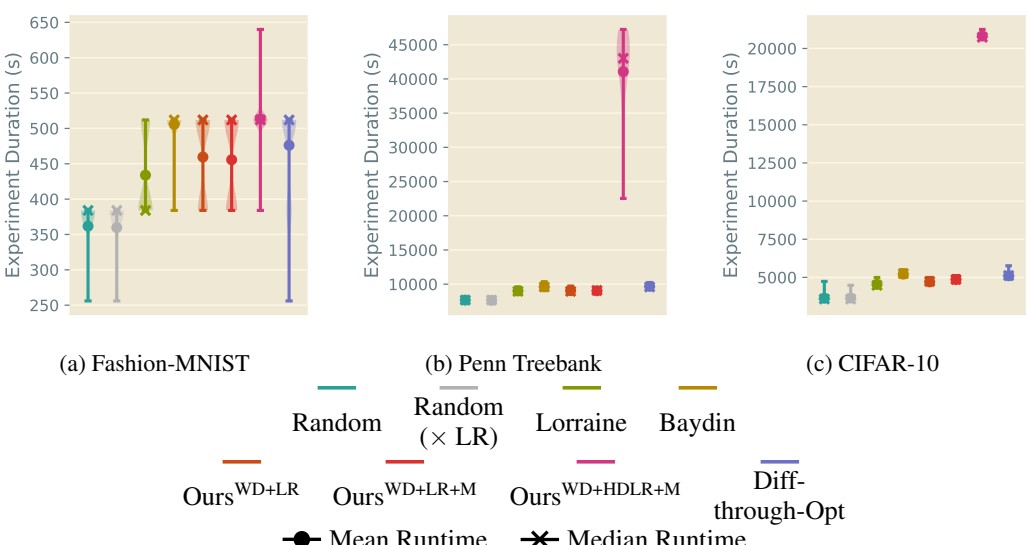

Figure 14: Single-pass runtime distributions for our larger-scale experiments. Worst-case complexities are dominated by second derivative computations, but remain competitive with conventional multi-pass HPO techniques, which scale our Random results by the number of configurations sampled.

For our larger-scale experiments, we illustrate comparative runtimes in Figure 14. More complex model architectures, with more learnable parameters, suffer a greater computational cost from taking second derivatives with respect to each model weight, which appears more prominently in our Penn Treebank and CIFAR-10 experiments. However, our extension of Lorraine et al. (2020)'s method to additional hyperparameters doesn't dramatically increase runtimes: second derivative computations themselves are responsible for most added complexity. In all three cases, our scalar algorithms have runtimes within that of naïvely training two fixed hyperparameter initialisations (such as those considered by *Random*), comparing extremely favourably to HPO methods requiring repeated retraining. Naturally, the massive increase in the number of hyperparameters optimised by *Ours*$^{WD+HDLR+M}$ causes a prominent jump in runtimes, but even this algorithm remains within a practical factor of the *Random* baseline.

### B.6.2 BATCH NORMALISATION AND FASHION-MNIST

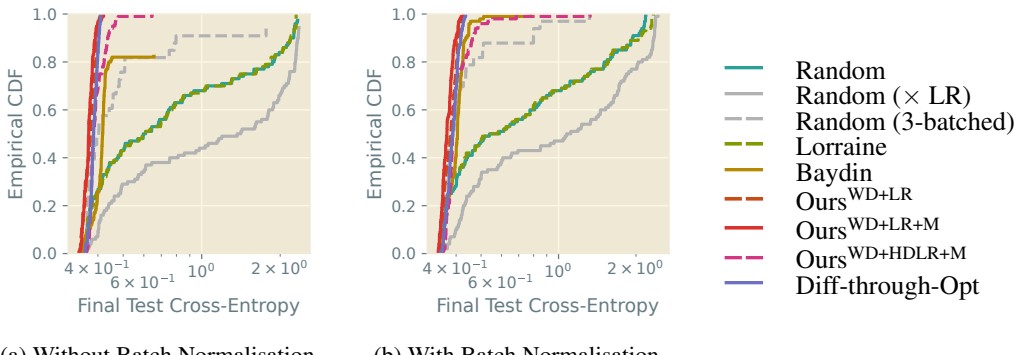

(a) Without Batch Normalisation      (b) With Batch Normalisation

Figure 15: Comparing CDFs of final test cross-entropy from our Fashion-MNIST experiments of Section 4.2: (a) as before, without batch normalisation; (b) with batch normalisation in our simple MLP model.

Table 9: Quantitative analysis of final test cross-entropies from Figure 15 — investigating the use of batch normalisation on Fashion-MNIST. Bold values are the lowest in class.

| Method | Without Batch Normalisation | | | With Batch Normalisation | | |
|---|---|---|---|---|---|---|
| | Mean | Median | Best | Mean | Median | Best |
| Random | $0.92 \pm 0.07$ | $0.60 \pm 0.08$ | $0.340$ | $0.88 \pm 0.06$ | $0.6 \pm 0.1$ | $\mathbf{0.333}$ |
| Random ($\times$ LR) | $1.29 \pm 0.08$ | $1.2 \pm 0.3$ | $0.339$ | $1.21 \pm 0.08$ | $1.1 \pm 0.2$ | $0.338$ |
| Random (3-batched) | $0.8 \pm 0.1$ | $0.49 \pm 0.09$ | $0.349$ | $0.42 \pm 0.02$ | $\mathbf{0.37} \pm 0.01$ | $0.340$ |
| Lorraine | $0.95 \pm 0.07$ | $0.61 \pm 0.08$ | $0.343$ | $0.89 \pm 0.06$ | $0.58 \pm 0.09$ | $0.343$ |
| Baydin | $0.409 \pm 0.004$ | $0.413 \pm 0.002$ | $0.337$ | $0.404 \pm 0.005$ | $0.404 \pm 0.002$ | $0.335$ |
| Ours$^{\text{WD+LR}}$ | $0.373 \pm 0.002$ | $0.375 \pm 0.003$ | $\mathbf{0.336}$ | $0.380 \pm 0.002$ | $0.381 \pm 0.004$ | $0.338$ |
| Ours$^{\text{WD+LR+M}}$ | $\mathbf{0.369} \pm 0.002$ | $\mathbf{0.369} \pm 0.001$ | $0.339$ | $\mathbf{0.371} \pm 0.002$ | $\mathbf{0.368} \pm 0.003$ | $0.337$ |
| Ours$^{\text{WD+HDLR+M}}$ | $0.398 \pm 0.004$ | $0.386 \pm 0.003$ | $0.350$ | $0.41 \pm 0.01$ | $0.388 \pm 0.002$ | $0.348$ |
| Diff-through-Opt | $0.386 \pm 0.001$ | $0.385 \pm 0.002$ | $0.355$ | $0.388 \pm 0.002$ | $0.388 \pm 0.004$ | $0.348$ |

As batch normalisation rescales the parameter gradients for each training step, it is capable of further influencing how final performance is governed by learning rates and their variations. To study this effect, we repeat the Fashion-MNIST experiments described in Section 4.2 using batch normalisation in our simple MLP model. Our results are plotted alongside the non-batch normalised CDFs in Figure 15, with averages summarised in Table 9.

Subjectively, performance does not seem to dramatically change when batch normalisation is invoked, with CDFs having broadly the same shape. Considering means and medians as our figures of merit (Table 9) suggests the same general pattern: generally, our algorithms benefit slightly from the absence of batch normalisation, and other algorithms benefit slightly from its presence, but there is not a unanimous trend. We conclude that, in the context of this work, batch normalisation does not substantially improve or hinder performance.

## B.7 EFFECT OF SHORT-HORIZON BIAS

Attempts to quantify the effect of short-horizon bias in our experiments are frustrated by the need to perform computationally-intensive longer-horizon experiments, which has limited the scope of evaluations we can perform. However, we are able to study the potential improvements of longer horizons in three experimental settings.

### B.7.1 TRUNCATED PROBLEM, FULL HORIZON

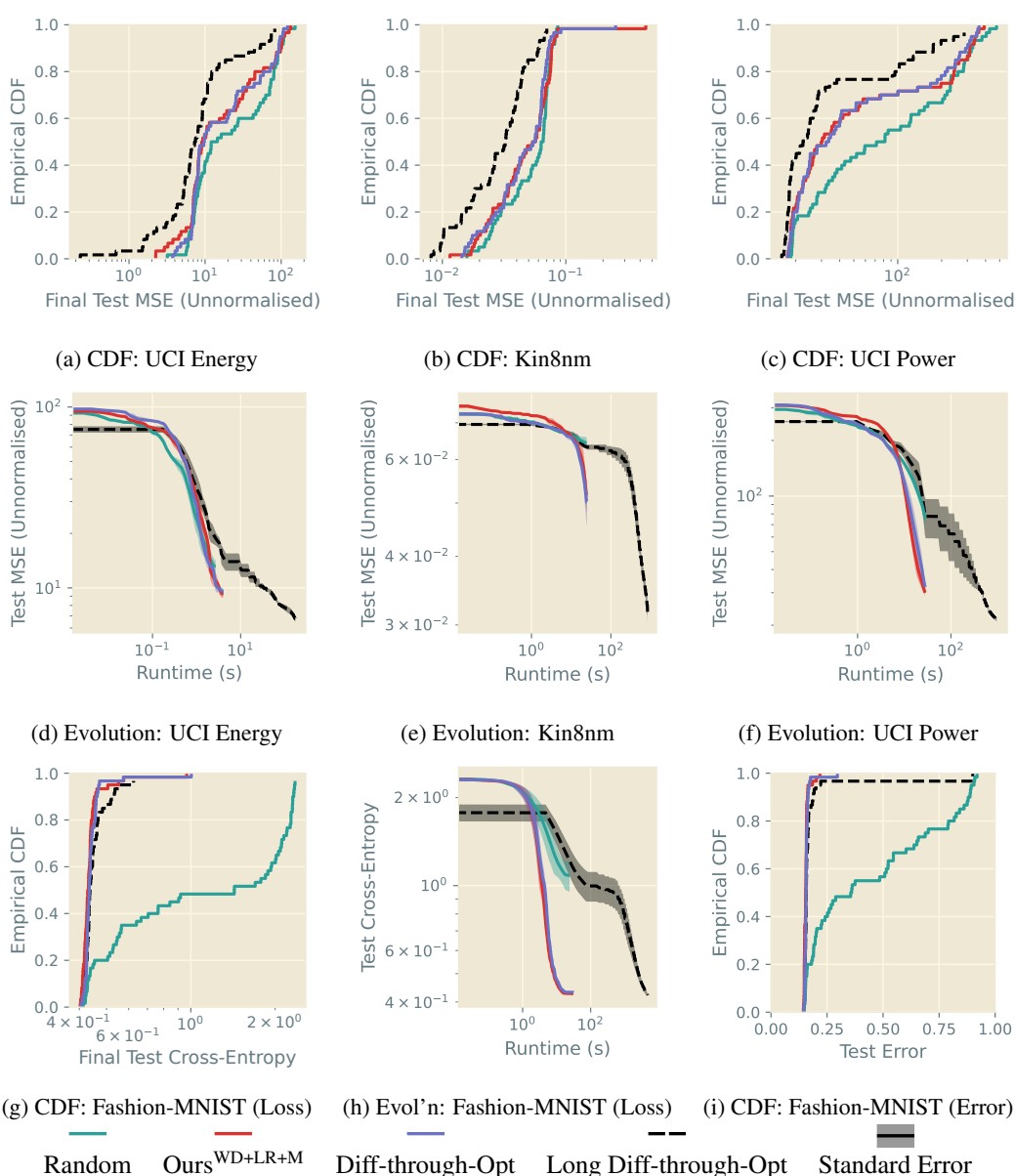

(a) CDF: UCI Energy     (b) CDF: Kin8nm     (c) CDF: UCI Power

(d) Evolution: UCI Energy     (e) Evolution: Kin8nm     (f) Evolution: UCI Power

(g) CDF: Fashion-MNIST (Loss)     (h) Evol'n: Fashion-MNIST (Loss)     (i) CDF: Fashion-MNIST (Error)

Random    Ours$^{\text{WD+LR+M}}$    Diff-through-Opt    Long Diff-through-Opt    Standard Error

Figure 16: CDFs of final test losses and evolutions of bootstrapped median losses from 60 random initialisations on a shorter problem of 200 (UCI and Kin8nm) / 1000 (Fashion-MNIST) network weight updates, featuring full-horizon *Long Diff-through-Opt* as described in Appendix B.7.1.

Firstly, we consider the four study datasets UCI Energy/Power, Kin8nm and Fashion-MNIST, shortening training so it only lasts for 200 (UCI and Kin8nm) / 1000 (Fashion-MNIST) network weight update steps. This shorter training episode means we cannot compare to our results in Section 4, but

Table 10: Quantitative analysis of UCI and Kin8nm final test performance from Figure 16, including full-horizon *Long Diff-through-Opt* on shorter problems as described in Appendix B.7.1. Bold values are the lowest in class.

| Method | UCI Energy | | | Kin8nm ($\times 1\,000$) | | | UCI Power | | |
|---|---|---|---|---|---|---|---|---|---|
| | Mean | Median | Best | Mean | Median | Best | Mean | Median | Best |
| Random | $40 \pm 5$ | $17 \pm 9$ | 3.19 | $56 \pm 3$ | $63 \pm 4$ | 15.8 | $130 \pm 20$ | $80 \pm 30$ | 18.5 |
| Ours$^{WD+LR+M}$ | $30 \pm 5$ | $10 \pm 3$ | 2.24 | $57 \pm 7$ | $54 \pm 7$ | 11.5 | $100 \pm 20$ | $31 \pm 6$ | 17.6 |
| Diff-through-Opt | $30 \pm 4$ | $10 \pm 3$ | 3.76 | $52 \pm 4$ | $53 \pm 6$ | 12.9 | $100 \pm 10$ | $33 \pm 6$ | 17.6 |
| Long Diff-through-Opt | $\mathbf{14} \pm 3$ | $\mathbf{7.3} \pm 0.9$ | **0.206** | $\mathbf{32} \pm 2$ | $\mathbf{31} \pm 3$ | **7.70** | $\mathbf{49} \pm 8$ | $\mathbf{22} \pm 2$ | **15.9** |

Table 11: Quantitative analysis of Fashion-MNIST final test performance from Figure 16, including full-horizon *Long Diff-through-Opt* on a shorter problem as described in Appendix B.7.1. Bold values are the lowest in class.

| Method | Final Test Cross-Entropy | | | Final Test Error | | |
|---|---|---|---|---|---|---|
| | Mean | Median | Best | Mean | Median | Best |
| Random | $1.3 \pm 0.1$ | $1.2 \pm 0.4$ | 0.408 | $0.45 \pm 0.04$ | $0.35 \pm 0.09$ | 0.146 |
| Ours$^{WD+LR+M}$ | $\mathbf{0.443} \pm 0.009$ | $\mathbf{0.430} \pm 0.003$ | 0.401 | $\mathbf{0.156} \pm 0.002$ | $\mathbf{0.155} \pm 0.001$ | 0.142 |
| Diff-through-Opt | $\mathbf{0.45} \pm 0.01$ | $0.434 \pm 0.002$ | 0.406 | $0.159 \pm 0.002$ | $0.1565 \pm 0.0007$ | 0.145 |
| Long Diff-through-Opt | $0.450 \pm 0.005$ | $0.436 \pm 0.003$ | **0.398** | $0.19 \pm 0.02$ | $0.1574 \pm 0.0009$ | **0.140** |

does permit full-horizon differentiation-through-optimisation. *Long Diff-through-Opt* does exactly this, performing 30 (UCI and Kin8nm) / 50 (Fashion-MNIST) hyperparameter updates using the maximum look-back horizon and restarting training with the new hyperparameters after each update. In this sense, we use 30 or 50 times as much computation as in our other algorithms (here: *Random*, *Ours$^{WD+LR+M}$* and *Diff-through-Opt*). The latter update their hyperparameters at the same intervals as before, giving 20 (UCI and Kin8nm) / 100 (Fashion-MNIST) updates overall. We take 60 random hyperparameter initialisations and run each of these four methods on them.

Our results are shown in Figure 16 and Tables 10 and 11. They show a stark contrast between datasets — on UCI and Kin8nm trials, *Long Diff-through-Opt* manages, as expected, to achieve substantially improved final performance over its short-horizon competition, but on Fashion-MNIST, performance is much more similar between HPO algorithms. Wu et al. (2018) intuit short-horizon bias by describing the simple setting of a quadratic with different curvature magnitudes in each dimension, and we expect the simpler UCI/Kin8nm datasets to match this clean depiction of the optimisation space more closely than Fashion-MNIST. Thus, we speculate the space of non-trivial datasets, such as Fashion-MNIST, is sufficiently complicated that merely eliminating short-horizon bias is insufficient to prevent the optimiser from getting stuck in similarly-performant local optima.

### B.7.2 FASHION-MNIST, MEDIUM HORIZON

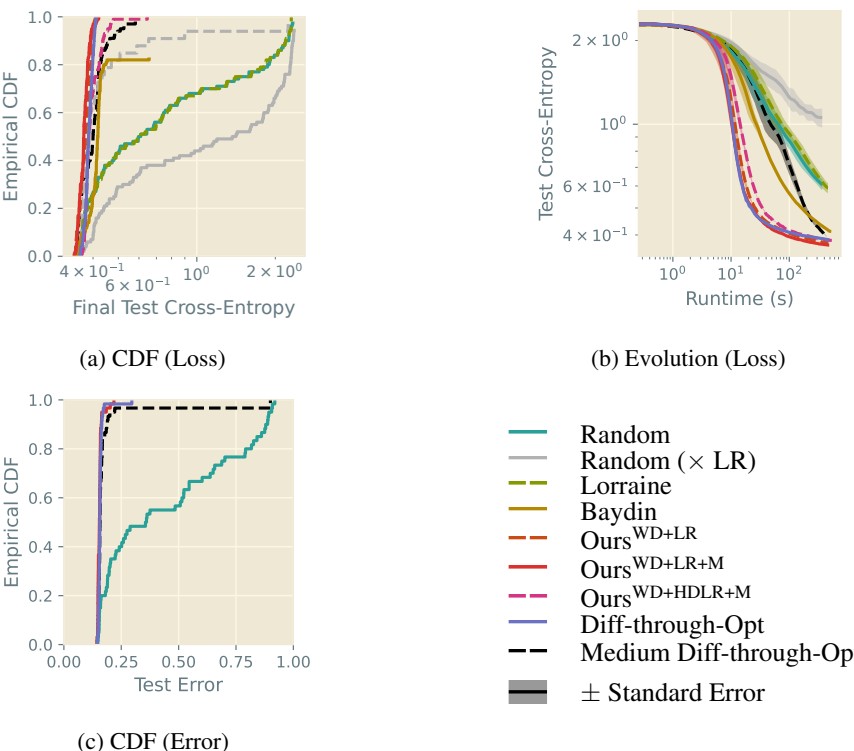

(a) CDF (Loss)

(b) Evolution (Loss)

(c) CDF (Error)

Figure 17: CDFs of final test losses and evolutions of bootstrapped median losses from 100 random initialisations on Fashion-MNIST, featuring *Medium Diff-through-Opt* with a 200-step look-back horizon as described in Appendix B.7.2.

Table 12: Quantitative analysis of Fashion-MNIST final test performance from Figure 17, including *Medium Diff-through-Opt* with a 200-step look-back horizon as described in Appendix B.7.2. Bold values are the lowest in class.

| Method | Final Test Cross-Entropy | | | Final Test Error | | |
|---|---|---|---|---|---|---|
| | Mean | Median | Best | Mean | Median | Best |
| Random | 0.92 ± 0.07 | 0.60 ± 0.08 | 0.340 | 0.31 ± 0.02 | 0.21 ± 0.02 | 0.116 |
| Random (× LR) | 1.29 ± 0.08 | 1.2 ± 0.3 | 0.339 | 0.46 ± 0.03 | 0.37 ± 0.06 | **0.115** |
| Random (3-batched) | 0.8 ± 0.1 | 0.49 ± 0.09 | 0.349 | 0.20 ± 0.03 | 0.141 ± 0.009 | 0.119 |
| Lorraine | 0.95 ± 0.07 | 0.61 ± 0.08 | 0.343 | 0.32 ± 0.02 | 0.22 ± 0.03 | 0.120 |
| Baydin | 0.409 ± 0.004 | 0.413 ± 0.002 | 0.337 | 0.27 ± 0.03 | 0.1499 ± 0.0008 | 0.117 |
| Ours$^{WD+LR}$ | 0.373 ± 0.002 | 0.375 ± 0.003 | **0.336** | 0.1329 ± 0.0007 | 0.133 ± 0.001 | 0.118 |
| Ours$^{WD+LR+M}$ | **0.369** ± 0.002 | **0.369** ± 0.001 | 0.339 | **0.1318** ± 0.0006 | **0.1316** ± 0.0006 | 0.119 |
| Ours$^{WD+HDLR+M}$ | 0.398 ± 0.004 | 0.386 ± 0.003 | 0.350 | 0.143 ± 0.001 | 0.1384 ± 0.0008 | 0.122 |
| Diff-through-Opt | 0.386 ± 0.001 | 0.385 ± 0.002 | 0.355 | 0.1376 ± 0.0005 | 0.1372 ± 0.0008 | 0.125 |
| Medium Diff-through-Opt | 0.400 ± 0.005 | 0.399 ± 0.004 | 0.338 | 0.16 ± 0.01 | 0.141 ± 0.002 | 0.116 |

Next, we note our Fashion-MNIST experiments of Section 4 run for 10 000 network weight steps, which can accommodate a longer look-back horizon for the purposes of our study. We introduce a new algorithm, *Medium Diff-through-Opt*, which is identical to our previous Fashion-MNIST *Diff-through-Opt* configuration, except the look-back horizon is now 200 steps, and the number of hyperparameter updates reduced to 50 to keep the same computational burden. This setting explores the trade-off between hypergradient accuracy and number of hypergradients computed. We apply this new configuration to the same 100 iterations as before, plotting combined results in Figure 17 and Table 12.

Again, we see that Fashion-MNIST seems resilient to the short-horizon bias, with this medium-horizon configuration unable to substantially improve on our existing results. Naturally, since *Medium Diff-through-Opt* performs fewer hyperparameter updates, its performance may suffer if the hyperparameters have not converged, which we expect to be the likely cause here. However, it is reassuring that our methods are capable of performing well despite their short-horizon foundations.

### B.7.3 UCI ENERGY, FULL HORIZON

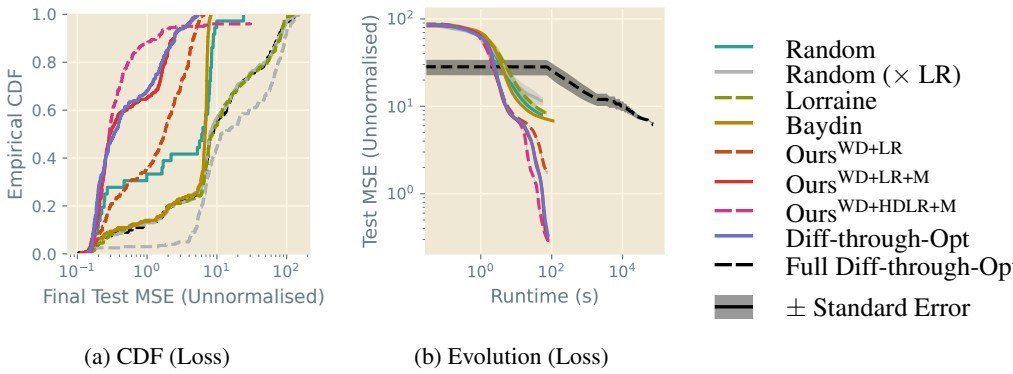

(a) CDF (Loss)      (b) Evolution (Loss)

Figure 18: CDF of final test losses and evolution of bootstrapped median losses from 200 random initialisations on UCI Energy, featuring 36 initialisations of *Full Diff-through-Opt* with a 4 000-step look-back horizon as described in Appendix B.7.3.

Table 13: Quantitative analysis of UCI Energy final test performance from Figure 18, including *Full Diff-through-Opt* with a 4 000-step look-back horizon as described in Appendix B.7.3. Bold values are the lowest in class.

| Method | Final Test MSE | | | | | |
|---|---|---|---|---|---|
| | Mean | | Median | | Best |
| Random | 24 | $\pm 2$ | 8.3 | $\pm 0.7$ | 0.124 |
| Random ($\times$ LR) | 35 | $\pm 3$ | 13 | $\pm 3$ | 0.113 |
| Random (3-batched) | 5 | $\pm 1$ | 2 | $\pm 1$ | **0.104** |
| Lorraine | 24 | $\pm 2$ | 8.7 | $\pm 0.9$ | 0.145 |
| Baydin | 5.5 | $\pm 0.2$ | 6.8 | $\pm 0.1$ | 0.144 |
| Ours$^{\text{WD+LR}}$ | 2.1 | $\pm 0.1$ | 1.8 | $\pm 0.2$ | 0.130 |
| Ours$^{\text{WD+LR+M}}$ | 0.96 | $\pm 0.08$ | 0.30 | $\pm 0.03$ | 0.112 |
| Ours$^{\text{WD+HDLR+M}}$ | **0.6** | $\pm 0.2$ | **0.28** | $\pm 0.01$ | 0.139 |
| Diff-through-Opt | 0.93 | $\pm 0.08$ | 0.34 | $\pm 0.04$ | 0.132 |
| Full Diff-through-Opt | 5.2 | $\pm 0.8$ | 6 | $\pm 2$ | 0.138 |

Finally, we consider UCI Energy, whose size is more amenable to full-horizon studies. Our new setting here, *Full Diff-through-Opt*, extends the *Diff-through-Opt* configuration from Section 4 by using a full look-back horizon of 4 000 steps, restarting training after each hyperparameter update and performing 30 such updates. This permits a full-horizon study on the same problem as before, using a similar computational footprint to our Bayesian Optimisation trials of Appendix B.5 (that is, 30 times the cost of our original experiments). We consider 36 random initialisations for this algorithm, and combine the results with our original UCI Energy experiments, showing both in Figure 18 and Table 13.

Curiously, while *Full Diff-through-Opt* outclasses naïve *Random* approaches, it is very uncompetitive with its short-horizon counterparts, achieving much higher average final losses (the larger variance being, of course, expected due to the smaller number of repetitions). The principal difference between *Full Diff-through-Opt* and *Diff-through-Opt*, apart from the different look-back distances, is the number of hyperparameter updates performed, since these are far more expensive in the former case. While more hyperparameter updates would likely cause *Full Diff-through-Opt*'s performance to improve, the computational impact would be massive, with such a setting unlikely to be desired in practical applications. Again, we conclude that it is simplistic to claim short-horizon bias renders an algorithm ill-suited to HPO — in reality, it is one of many factors worthy of consideration in the selection of an approach. In this case, the larger number of updates we are able to perform by sacrificing horizon length leads to favourable performance improvements, even though this optimisation is technically 'biased' as a result.

Table 14: Final test MSEs after training UCI Energy as in Table 1, with dataset splits modified to the indicated training/validation/test proportions. Uncertainties are standard errors, and the values may be directly compared to Table 1.

| Method | Default (80%/10%/10%) | | | 67.5%/22.5%/10% | | | 56.25%/33.75%/10% | | | 45%/45%/10% | | |
|---|---|---|---|---|---|---|---|---|---|---|---|---|
| | Mean | Median | Best | Mean | Median | Best | Mean | Median | Best | Mean | Median | Best |
| Ours$^{WD+LR+M}$ | $0.96_{\pm 0.08}$ | $0.30_{\pm 0.03}$ | 0.112 | $0.81_{\pm 0.09}$ | $0.34_{\pm 0.03}$ | 0.127 | $1.01_{\pm 0.08}$ | $0.47_{\pm 0.05}$ | 0.184 | $1.04_{\pm 0.09}$ | $0.52_{\pm 0.03}$ | 0.204 |
| Ours$^{WD+HDLR+M}$ | $0.6_{\pm 0.2}$ | $0.28_{\pm 0.01}$ | 0.139 | $0.35_{\pm 0.05}$ | $0.234_{\pm 0.005}$ | 0.143 | $0.8_{\pm 0.5}$ | $0.260_{\pm 0.006}$ | 0.160 | $0.9_{\pm 0.5}$ | $0.33_{\pm 0.01}$ | 0.181 |

## B.8 UCI ENERGY: FURTHER ANALYSIS

In light of our main results, we perform several additional experiments to investigate particular aspects of our algorithm. These are conducted on the UCI Energy dataset, using the same configuration as in Section 4.1 except where otherwise noted.

### B.8.1 VALIDATION DATASET SIZE

Noting that results for $Ours^{WD+HDLR+M}$ did not always outperform $Ours^{WD+LR+M}$ as might have been expected, and considering the risk of overfitting to validation data, we consider the role the validation dataset might have to play on final test performance. The conventional training/validation/test dataset split sizes we use throughout this paper produce a training set which is substantially larger than validation or test. In $Our^{WD+HDLR+M}$ case, we have approximately the same number of hyperparameters as model weights, which is not the case in classical training. Thus, we posit that choosing a larger validation dataset may permit improved hyperparameter tuning, addressing this performance gap.

In Table 14 we show results over 200 random initialisations, with bootstrapped error bars computed as in Section 4.1. We fix the test set size at our default 10% of the available data (per Gal & Ghahramani (2016)), and vary the balance between training and validation data as shown.

Both the scalar and high-dimensional settings clearly benefit from a larger validation dataset, with a 67.5%/22.5%/10% split significantly improving performance. But $Ours^{WD+HDLR+M}$ seems to exhibit less reliable, higher-variance behaviour at even larger validation splits, with $Ours^{WD+LR+M}$ also suffering worse performance in these settings. The differences between best-case performance in these algorithms suggest there is yet more to be understood about $Our^{WD+HDLR+M}$ setting. However, it is clear that with larger numbers of hyperparameters, we must take care to balance the competing demands for training and validation data.

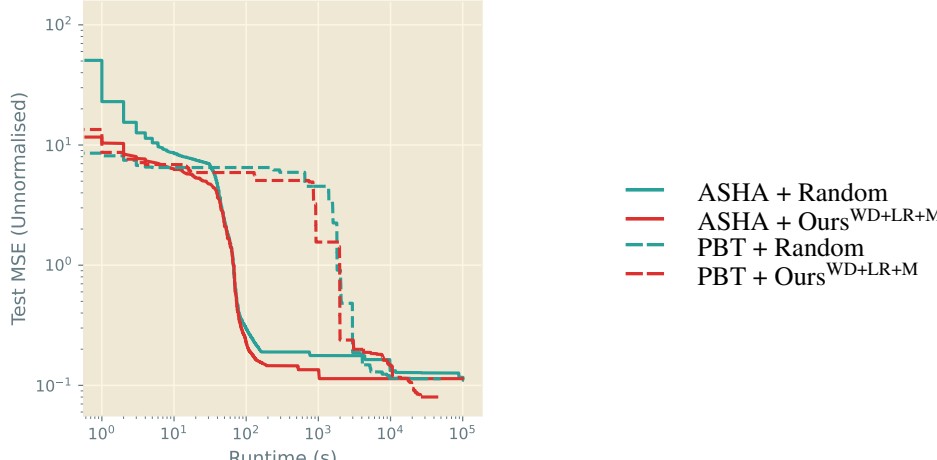

Figure 19: Evolution of incumbent best test MSE with time, during training of UCI Energy on ASHA (114 286 initialisations) and PBT (200 training paths), with the threshold for 'fully-trained' set at 4 000 iterations. Note the logarithmic axes.

Table 15: Extended reprise of Table 1: final test MSEs after training UCI Energy for 4 000 iterations, averaged over 200 initialisations. Also shown are statistics for the initialisations which survive to full training on ASHA (out of 114 286 considered) and the final models produced by PBT (out of 200 training paths). Uncertainties are standard errors; bold values lie in the error bars of the best algorithm.

| Method | UCI Energy | | | | | |
|---|---|---|---|---|---|
| | Mean | | Median | | Best |
| Random | 24 | $\pm 2$ | 8.3 | $\pm 0.7$ | 0.124 |
| Random ($\times$ LR) | 35 | $\pm 3$ | 13 | $\pm 3$ | 0.113 |
| Random (3-batched) | 5 | $\pm 1$ | 2 | $\pm 1$ | 0.104 |
| Lorraine | 24 | $\pm 2$ | 8.7 | $\pm 0.9$ | 0.145 |
| Baydin | 5.5 | $\pm 0.2$ | 6.8 | $\pm 0.1$ | 0.144 |
| Ours$^{WD+LR}$ | 2.1 | $\pm 0.1$ | 1.8 | $\pm 0.2$ | 0.130 |
| Ours$^{WD+LR+M}$ | 0.96 | $\pm 0.08$ | 0.30 | $\pm 0.03$ | 0.112 |
| Ours$^{WD+HDLR+M}$ | 0.6 | $\pm 0.2$ | 0.28 | $\pm 0.01$ | 0.139 |
| Diff-through-Opt | 0.93 | $\pm 0.08$ | 0.34 | $\pm 0.04$ | 0.132 |
| ASHA + Random | 0.29 | $\pm 0.09$ | 0.191 | $\pm 0.004$ | 0.131 |
| ASHA + Ours$^{WD+LR+M}$ | 0.160 | $\pm 0.004$ | 0.162 | $\pm 0.003$ | 0.104 |
| PBT + Random | 0.18 | $\pm 0.03$ | 0.1328 | $\pm 0.0007$ | **0.0934** |
| PBT + Ours$^{WD+LR+M}$ | **0.134** | $\pm 0.005$ | **0.126** | $\pm 0.003$ | 0.119 |

### B.8.2 ASYNCHRONOUS HYPERBAND AND POPULATION-BASED TRAINING

Finally, we consider two further non-gradient-based HPO strategies: Asynchronous Hyperband (ASHA; Li et al. (2020b)) and Population-Based Training (PBT; Jaderberg et al. (2017)). Both approaches exploit parallel computation to consider a population of hyperparameter selections, and the performance of each individual selection during training is used to focus computational effort on the most promising trials. Broadly speaking, ASHA does this by terminating poorly-performing trials early, while PBT locally perturbs trials or replaces them entirely with better-performing checkpoints. In this sense, ASHA and PBT are orthogonal to the type of algorithm we propose in this paper, which seeks to improve the performance obtained by single initialisations trained in isolation. However, our algorithm can readily be combined with ASHA or PBT: by training each proposed hyperparameter choice using our algorithm instead of fixing the hyperparameters as is conventional, we exploit our method's benefits for each individual run, thus providing ASHA/PBT with a stronger population of

trials from which to make optimisation decisions. We demonstrate this synergy in the experiments of this section.

Throughout, we make use of the Ray Tune (Liaw et al., 2018) library's implementation of these strategies, utilising the default settings unless noted, and consider the UCI Energy dataset using the same setup as in Section 4.1. For the *Random* setting, we do not combine the training and validation datasets, instead training with the former and reserving the latter for the evaluation of different initialisations by ASHA/PBT. For $Our^{WD+LR+M}$ setting, the validation set is used both for gradient-based hyperparameter updates during training and for configuration evaluation by ASHA/PBT. Otherwise, both settings are the same as in Section 4, and again we quote final losses on a held-out test set, with 1 000 full-dataset bootstrap samples used to compute standard errors. We also leave the hyperparameter search space unchanged from before (see Section B.3).

**Asynchronous Hyperband**   For ASHA, we seek to match the $4\,000 \times 200$ weight update steps performed in Section 4.1, with 4 000 steps still required for full training. Under the default settings of Ray Tune (no grace period, reduction factor 4, one bracket), we obtain computational budgets of $(1, 4, 16, 64, 256, 1\,024, 4\,000)$ iterations, with the number of initialisations surviving to each budget step correspondingly decreasing by a factor of 4 each time. Non-parallelised Hyperband (Li et al., 2017) uses the same total computational effort for each budget step; applying this fact to the smallest computational budget gives that we must begin with $\frac{4\,000 \times 200}{7 \times 1} \approx 114\,286$ initialisations for computational comparability. While the asynchronous modifications of ASHA introduce additional stochasticity, we assume the expected workload to be the same as the non-parallelised case. We run eight workers in parallel to mimic the experimental setting of our earlier studies.

From the results in Table 15, we see our hypothesis largely confirmed. By virtue of curating the population of initialisations, rather than considering each one in isolation, ASHA trivially gives improved average performance over all earlier methods. However, combining ASHA with $Ours^{WD+LR+M}$ yields a significantly more performant population of trials on average, which leads to low-valued, low-variance mean and median final test performance. In both ASHA settings, the best final losses in the resulting populations are towards the lower end of those found in our previous results, but do not beat the especially small loss found by *Random (3-batched)*. From this, we suppose that the methods we consider are nearing the performance limits of the UCI Energy dataset on this model when trained in this way. But regardless, we clearly see that deploying our algorithm in a particular context tends to improve average-case performance, indicating a positive influence on training.

**Population-Based Training**   For PBT, we again attempt to match the earlier computational burden by training 200 initialisations for 4 000 iterations using 8-way parallelism, acknowledging that PBT's checkpoint restoration will cause most runs to perform more than 4 000 weight update steps. We set the perturbation interval to 100 steps, so each initialisation has 40 opportunities to be mutated or reset by PBT. In *PBT + Ours$^{WD+LR+M}$*, it is unclear how to accommodate PBT's hyperparameter mutations, as our algorithm will have updated the hyperparameters PBT associates with each trial. Where we detect PBT has mutated the hyperparameters, we compute the ratio between the new and old values as seen by PBT, then apply that same ratio to mutate the true current value of the hyperparameters.

Our PBT results (Table 15) largely continue the pattern we have discussed. Combining our algorithm with PBT achieves substantial improvements in mean and median performance, giving the lowest values seen in any of our experiments on UCI Energy. PBT uses additional internal machinery to manage the population of hyperparameter initialisations, and we found this caused the total training time to considerably increase over the non-population-based methods of Section 4.1, so the performance improvements we see here are perhaps unsurprising (see also Figure 19 and our discussion of runtimes below). However, in situations where we have already committed to the additional cost of PBT, replacing the fixed-hyperparameter internal training routine with our algorithm clearly results in a superior population of models, improving the quality of the models available for downstream applications. Note that the quality of the search over hyperparameter initialisations affects our overall best loss found, so some variation in the Best column of Table 15 is to be expected.

**Runtime Analysis**   Figure 19 shows a runtime analysis of these results on ASHA and PBT. We see clearly that the superior performance of $Ours^{WD+LR+M}$ in isolated training of specific hyperparameter initialisations causes ASHA to reach low losses faster. In PBT, there is little difference in performance

vs. runtime when *Random* or $Ours^{WD+LR+M}$ is used as the internal learning strategy. Subjectively, this is likely because the overhead of saving checkpoints and restarting from them dominated the overall cost of running PBT. In both cases, PBT is substantially slower than ASHA to reach the most performant losses. Note by comparison with Figure 11a that our methods alone are competitive with ASHA when we compare the evolution of incumbent best performance over time. Note also that ASHA and PBT do not have access to the intermediate test losses plotted in Figure 19, so the smallest test losses found are not necessarily achievable as an output of these methods.

## C  COMPLETE DERIVATION

Here, we reprise the derivations of Section 2 in more verbose form. Our subsections mirror those used previously.

### C.1  IMPLICIT FUNCTION THEOREM IN BILEVEL OPTIMISATION

Recall that we consider some model parameterised by parameters $\mathbf{w}$, which we train to minimise a training loss $\mathcal{L}_T$. Our training algorithm is governed by hyperparameters $\boldsymbol{\lambda}$, which we simultaneously train to minimise a validation loss $\mathcal{L}_V$. We target optimal $\mathbf{w}^*$ and $\boldsymbol{\lambda}^*$, defined by the *bilevel optimisation problem*

$$^{(a)}\ \boldsymbol{\lambda}^* = \arg\min_{\boldsymbol{\lambda}} \mathcal{L}_V(\boldsymbol{\lambda}, \mathbf{w}^*(\boldsymbol{\lambda})), \quad \text{such that} \quad ^{(b)}\ \mathbf{w}^*(\boldsymbol{\lambda}) = \arg\min_{\mathbf{w}} \mathcal{L}_T(\boldsymbol{\lambda}, \mathbf{w}). \quad \text{(1 revisited)}$$

(1b) simply reflects a standard model training procedure, with fixed $\boldsymbol{\lambda}$, so we need only consider how to choose the $\boldsymbol{\lambda}^*$ from (1a).

Suppose we seek a gradient-based hyperparameter optimisation method for its performance potential. We then require the total derivative of the validation objective $\mathcal{L}_V$ with respect to the hyperparameters $\boldsymbol{\lambda}$, which may be obtained by the chain rule:

$$\frac{\mathrm{d}\mathcal{L}_V}{\mathrm{d}\boldsymbol{\lambda}} = \frac{\partial \mathcal{L}_V}{\partial \boldsymbol{\lambda}} + \frac{\partial \mathcal{L}_V}{\partial \mathbf{w}^*} \frac{\partial \mathbf{w}^*}{\partial \boldsymbol{\lambda}}. \quad \text{(2 revisited)}$$

An exact expression for $\frac{\partial \mathbf{w}^*}{\partial \boldsymbol{\lambda}}$ can be obtained by applying Cauchy's Implicit Function Theorem (Theorem 1). Suppose that for some $\boldsymbol{\lambda}'$ we have identified a locally optimal $\mathbf{w}'$, and assume continuous differentiability of $\frac{\partial \mathcal{L}_T}{\partial \mathbf{w}}$. Then, there exists an open subset of hyperparameter space containing $\boldsymbol{\lambda}'$, over which we can define a function $\mathbf{w}^*(\boldsymbol{\lambda})$ relating locally optimal $\mathbf{w}$ for each $\boldsymbol{\lambda}$ by

$$\frac{\partial \mathbf{w}^*}{\partial \boldsymbol{\lambda}} = -\left(\frac{\partial^2 \mathcal{L}_T}{\partial \mathbf{w} \partial \mathbf{w}^{\mathsf{T}}}\right)^{-1} \frac{\partial^2 \mathcal{L}_T}{\partial \mathbf{w} \partial \boldsymbol{\lambda}^{\mathsf{T}}}. \quad \text{(3 revisited)}$$

Of course, the inverse Hessian term in (3) becomes rapidly intractable for models at realistic scale.

### C.2  APPROXIMATE BEST-RESPONSE DERIVATIVE

To proceed, we require an approximate expression for $\frac{\partial \mathbf{w}^*}{\partial \boldsymbol{\lambda}}$. Suppose model training proceeds iteratively by the rule

$$\mathbf{w}_i(\boldsymbol{\lambda}) = \mathbf{w}_{i-1}(\boldsymbol{\lambda}) - \mathbf{u}(\boldsymbol{\lambda}, \mathbf{w}_{i-1}(\boldsymbol{\lambda})), \quad \text{(4 revisited)}$$

where $i$ is the current iteration and $\mathbf{u}$ is an arbitrary function differentiable in both its arguments. Note that the trivial choice

$$\mathbf{u}(\boldsymbol{\lambda}, \mathbf{w}_{i-1}(\boldsymbol{\lambda})) = \mathbf{w}_{i-1}(\boldsymbol{\lambda}) - \mathbf{f}(\boldsymbol{\lambda}, \mathbf{w}_{i-1}(\boldsymbol{\lambda})) \quad (12)$$

allows arbitrary differentiable weight updates $\mathbf{f}$ to be subsumed by this framework.

Differentiating with respect to $\boldsymbol{\lambda}$ at the current hyperparameters $\boldsymbol{\lambda}'$, we have:

$$\left[\frac{\partial \mathbf{w}_i}{\partial \boldsymbol{\lambda}}\right]_{\boldsymbol{\lambda}'} = \left[\frac{\partial \mathbf{w}_{i-1}}{\partial \boldsymbol{\lambda}}\right]_{\boldsymbol{\lambda}'} - \left[\frac{\mathrm{d}\mathbf{u}}{\mathrm{d}\boldsymbol{\lambda}}\right]_{\boldsymbol{\lambda}', \mathbf{w}_{i-1}(\boldsymbol{\lambda}')} \quad (13)$$

$$= \left[\frac{\partial \mathbf{w}_{i-1}}{\partial \boldsymbol{\lambda}}\right]_{\boldsymbol{\lambda}'} - \left[\frac{\partial \mathbf{u}}{\partial \mathbf{w}} \frac{\partial \mathbf{w}_{i-1}}{\partial \boldsymbol{\lambda}} + \frac{\partial \mathbf{u}}{\partial \boldsymbol{\lambda}}\right]_{\boldsymbol{\lambda}', \mathbf{w}_{i-1}(\boldsymbol{\lambda}')} \quad (14)$$

$$= \left[\left(\mathbf{I} - \frac{\partial \mathbf{u}}{\partial \mathbf{w}}\right) \frac{\partial \mathbf{w}_{i-1}}{\partial \boldsymbol{\lambda}} - \frac{\partial \mathbf{u}}{\partial \boldsymbol{\lambda}}\right]_{\boldsymbol{\lambda}', \mathbf{w}_{i-1}(\boldsymbol{\lambda}')} \quad (15)$$

$$= -\left[\frac{\partial \mathbf{u}}{\partial \boldsymbol{\lambda}}\right]_{\boldsymbol{\lambda}', \mathbf{w}_{i-1}(\boldsymbol{\lambda}')} + \left[\mathbf{I} - \frac{\partial \mathbf{u}}{\partial \mathbf{w}}\right]_{\boldsymbol{\lambda}', \mathbf{w}_{i-1}(\boldsymbol{\lambda}')} \left[\left(\mathbf{I} - \frac{\partial \mathbf{u}}{\partial \mathbf{w}}\right) \frac{\partial \mathbf{w}_{i-2}}{\partial \boldsymbol{\lambda}} - \frac{\partial \mathbf{u}}{\partial \boldsymbol{\lambda}}\right]_{\boldsymbol{\lambda}', \mathbf{w}_{i-2}(\boldsymbol{\lambda}')} \quad (16)$$

$$= -\sum_{0 \leq j < i} \left(\left(\prod_{0 \leq k < j} \left[\mathbf{I} - \frac{\partial \mathbf{u}}{\partial \mathbf{w}}\right]_{\boldsymbol{\lambda}', \mathbf{w}_{i-1-k}(\boldsymbol{\lambda}')}\right) \left[\frac{\partial \mathbf{u}}{\partial \boldsymbol{\lambda}}\right]_{\boldsymbol{\lambda}', \mathbf{w}_{i-1-j}(\boldsymbol{\lambda}')}\right) \quad \text{(5 revisited)}$$

To combine terms in (5), we must make approximations such that all bracketed expressions are evaluated at the same $\boldsymbol{\lambda}, \mathbf{w}$. For network weights $\mathbf{w}$, we set $\mathbf{w}_0 = \mathbf{w}_i = \mathbf{w}^*$ for all iterations $i$, effectively assuming we have been at the optimal $\mathbf{w}$ for some time; when we come to consider finite truncations of the summation, this approximation will be especially appealing. With this approximation, we may collapse the product and write

$$\left[\frac{\partial \mathbf{w}_i}{\partial \boldsymbol{\lambda}}\right]_{\boldsymbol{\lambda}'} \approx -\sum_{0 \le j < i}\left[\left(\prod_{0 \le k < j}\left(\mathbf{I} - \frac{\partial \mathbf{u}}{\partial \mathbf{w}}\right)\right)\frac{\partial \mathbf{u}}{\partial \boldsymbol{\lambda}}\right]_{\boldsymbol{\lambda}', \mathbf{w}_i(\boldsymbol{\lambda}')} \tag{17}$$

$$= -\left[\sum_{0 \le j < i}\left(\mathbf{I} - \frac{\partial \mathbf{u}}{\partial \mathbf{w}}\right)^j \frac{\partial \mathbf{u}}{\partial \boldsymbol{\lambda}}\right]_{\boldsymbol{\lambda}', \mathbf{w}_i(\boldsymbol{\lambda}')}. \tag{6 revisited}$$

As $i$ increases, our training algorithm should converge towards a locally optimal $\mathbf{w}^*$, and the summation in (6) will accrue more terms. Instead, we reinterpret $i$ as specifying the finite number of terms we consider as an approximation to the whole sum. With this redefined $i$, we arrive at our approximate best-response Jacobian

$$\left[\frac{\partial \mathbf{w}^*}{\partial \boldsymbol{\lambda}}\right]_{\boldsymbol{\lambda}'} \approx -\left[\sum_{0 \le j < i}\left(\mathbf{I} - \frac{\partial \mathbf{u}}{\partial \mathbf{w}}\right)^j \frac{\partial \mathbf{u}}{\partial \boldsymbol{\lambda}}\right]_{\boldsymbol{\lambda}', \mathbf{w}^*(\boldsymbol{\lambda}')}. \tag{6 revisited}$$

### C.3 Convergence to Best-Response Derivative

To informally argue the convergence properties of our method, note that (6) is a truncated Neumann series:

$$\left[\frac{\partial \mathbf{w}_i}{\partial \boldsymbol{\lambda}}\right]_{\boldsymbol{\lambda}'} \approx -\left[\sum_{0 \le j < i}\left(\mathbf{I} - \frac{\partial \mathbf{u}}{\partial \mathbf{w}}\right)^j \frac{\partial \mathbf{u}}{\partial \boldsymbol{\lambda}}\right]_{\boldsymbol{\lambda}', \mathbf{w}^*(\boldsymbol{\lambda}')}. \tag{18}$$

Analogously to its scalar equivalent — the geometric series — this series converges as $i \to \infty$ if the multiplicative term $(\mathbf{I} - \frac{\partial \mathbf{u}}{\partial \mathbf{w}})$ is 'contractive' in some sense. For our purposes, it suffices to assume a Banach space and, for the operator norm $\|\cdot\|$, that $\|\mathbf{I} - \frac{\partial \mathbf{u}}{\partial \mathbf{w}}\| < 1$. In this case, we can directly apply the closed-form limit expression for the Neumann series:

$$\lim_{i \to \infty}\left[\frac{\partial \mathbf{w}_i}{\partial \boldsymbol{\lambda}}\right]_{\boldsymbol{\lambda}'} \approx -\lim_{i \to \infty}\left[\sum_{0 \le j < i}\left(\mathbf{I} - \frac{\partial \mathbf{u}}{\partial \mathbf{w}}\right)^j \frac{\partial \mathbf{u}}{\partial \boldsymbol{\lambda}}\right]_{\boldsymbol{\lambda}', \mathbf{w}^*(\boldsymbol{\lambda}')} \tag{19}$$

$$= -\left[\left(\mathbf{I} - \left(\mathbf{I} - \frac{\partial \mathbf{u}}{\partial \mathbf{w}}\right)\right)^{-1} \frac{\partial \mathbf{u}}{\partial \boldsymbol{\lambda}}\right]_{\boldsymbol{\lambda}', \mathbf{w}^*(\boldsymbol{\lambda}')} \tag{20}$$

$$= -\left[\left(\frac{\partial \mathbf{u}}{\partial \mathbf{w}}\right)^{-1} \frac{\partial \mathbf{u}}{\partial \boldsymbol{\lambda}}\right]_{\boldsymbol{\lambda}', \mathbf{w}^*(\boldsymbol{\lambda}')}. \tag{7 revisited}$$

Note that (7) is exactly the result of the Implicit Function Theorem (3) applied to our more general weight update $\mathbf{u}$. Concretely, suppose for some $\boldsymbol{\lambda}''$ and $\mathbf{w}''$ that $\mathbf{u}(\boldsymbol{\lambda}'', \mathbf{w}'') = \mathbf{0}$; in words, that our training process does not update the network weights at this point in weight-hyperparameter space, and assume $\mathbf{u}$ is continuously differentiable with invertible Jacobian. Then, over some subset of hyperparameter space containing $\boldsymbol{\lambda}''$, we can say: $\mathbf{w}^*(\boldsymbol{\lambda})$ exists, we have $\mathbf{u}(\boldsymbol{\lambda}, \mathbf{w}^*(\boldsymbol{\lambda})) = \mathbf{0}$ throughout the subspace and $\frac{\partial \mathbf{w}^*}{\partial \boldsymbol{\lambda}}$ is given by (7) exactly.

### C.4 Summary of Differences from Lorraine et al. (2020)

In their derivation, which ours closely parallels, Lorraine et al. (2020) replace (4) with a hard-coded SGD update step; in our notation, they fix $\mathbf{u}(\boldsymbol{\lambda}, \mathbf{w}) = \mathbf{u}_{\text{SGD}}(\boldsymbol{\lambda}, \mathbf{w}) = \eta \frac{\partial \mathcal{L}_T}{\partial \mathbf{w}}$. They then isolate the

learning rate $\eta$, insisting all hyperparameters $\boldsymbol{\lambda}$ must be dependent variables of the training loss $\mathcal{L}_T$, before further differentiating the training loss in their derivation. Consequently, their algorithm cannot optimise optimiser hyperparameters such as learning rates and momentum factors, since these come into play only *after* the training loss has been computed and differentiated. While a careful choice of transformed loss function may allow certain optimiser hyperparameters to be handled by the framework of Lorraine et al. (2020) (e.g. setting $\overline{\mathcal{L}} = \eta\mathcal{L}_T$ to incorporate a learning rate), this manual rederivation is far from trivial in more complex cases (e.g. to incorporate momentum).

Instead, we define a *general* weight update step (4), without assuming any particular form. We may then differentiate the general update $\mathbf{u}$ directly, encompassing any hyperparameters appearing within it but outside the training loss, which Lorraine et al. (2020) would miss. Crucially, this includes the hyperparameters of most gradient-based optimisers, including SGD. Thus, we lift Lorraine et al.'s results from expressions about $\frac{\partial\mathcal{L}_T}{\partial\mathbf{w}}$ to expressions about $\mathbf{u}$, then update hyperparameters inaccessible to the former. Concretely: we optimise weight decay coefficients, learning rates and momentum factors in settings where Lorraine et al. can only manage the weight decay alone.

