# OpenReview forum: "Scalable One-Pass Optimisation of High-Dimensional Weight-Update Hyperparameters by Implicit Differentiation"
_ICLR.cc/2022/Conference — ICLR 2022 Spotlight_

### Official Review · Reviewer_1R41 · 2021-10-24

**Correctness:** 4
**Technical Novelty And Significance:** 3
**Empirical Novelty And Significance:** 3
**Recommendation:** 8
**Confidence:** 3

**Main Review:**

**Clarity**

The paper is well written and generally easy to follow. It also includes useful visualizations, such as in Figure 1, that give the reader good intuition on the proposed derivation. It also fits the work well in the context of related work both on standard and one-pass HPO techniques (e.g., Lorraine et al. (2019)).

**Reproducibility**

Not only do the authors carefully provide details on the baselines, benchmarks, dataset sources, compute machines used and experimental setup, but they will also open source their code to reproduce the experiments upon acceptance. While the authors have linked to an anonymous repository, code is not visible yet. It would have been ideal to release that already, but apart from that the work meets a good bar in terms of reproducibility.

**Technical**

The work is well evaluated through experiments, which are performance on a large number of repetitions and illustrated with properly defined error bars. The proposed approximations are also theoretically grounded and converge to the ground truth best-response Jacobian as the size of the look-back window grows.

One weakness of the proposed approach is that it relies on a set of meta-hyperparameters, which the authors acknowledge. Ideally, their setting should be make automatic. In the absence of that, how would the author suggest setting them on a new problem? Is tuning on validation set required, or do they recommend some default values / can they give guidance on how to set them?

Related to that, I would have liked some clarifications on the following statement "As some settings habitually chose unstable high learning rates, we clip these to [10−10 , 1]" Why is that the case that unstable learning rates are used? Is this clipping a general recommendation?

Further, it was unclear to me why WD+HDLR+M would not perform well against the scalar baselines. The authors hint to learning rate regularisation for the high-dimensional dynamics setting, but some early results on that front would have further strengthened the paper.

**Minor**

A few of the listed references are pointing to arXiv when the respective paper was also published at a peer-reviewed venue. I recommend the authors to double-check those and update accordingly.



**Summary Of The Paper:**

The paper aims to accelerate HPO optimization by allowing for retraining when new hyperparameters are proposed, while being faster and more flexible that existing hypergradient-based one-pass methods. This results in a novel HPO one-pass technique that works with any continuous hyperparameter within a differentiable model weight update, such as learning rates. In a good range of experiments, the authors show that their proposal is faster than baselines.

**Summary Of The Review:**

I am inclined towards acceptance, only weakly so at this stage as guidance on setting the introduced meta-hyperparameters could be improved and the unexpected results obtained with WD+HDLR+M are lacking follow-up experiments. That said, this is overall solid work which is well-evaluated both theoretically and empirically.

---

> ### Author Response · Authors · 2021-11-22
> **Response to Reviewer 1R41**
>
> Thank you for taking the time to review our submission; we really
> appreciate your efforts. We’d like to address your comments and
> questions below.
>
> ## Meta-Hyperparameters and Clipping
>
> Our meta-hyperparameters were set following the previous work of
> Lorraine et al. (2019). Although we acknowledge our continued dependence
> on them is not ideal, previous work we cite on p6 (Franceschi et al.
> 2017, Franceschi et al. 2018 and Majumder et al. 2019) contains
> experimental results which give hope that it may not be vital to spend
> great computational effort in tuning them. In particular, Majumder et
> al. (2019) fix the best initial learning rate they found for each
> experimental setting, and repeat their optimisation with different
> choices of the hyper-learning rate used to learn the evolving learning
> rates, finding the final performance is not significantly affected by
> changes in the hyper-learning rate. In any case, our experimental
> results were all obtained with the same meta-hyperparameters, so we
> would suggest reusing these in new problems in the first instance.
>
> We introduced learning rate clipping as an ad-hoc defence to large
> learning rates which some settings were prone to selecting. While we
> don’t have an empirically-backed explanation for this, we note loss
> landscapes are known to contain substantial ‘flat’ regions with small
> gradients. With this in mind, we hypothesise our algorithms may be
> reaching such regions of the space, then studiously increasing the
> learning rate in an attempt to make more progress. If we then come
> across a valley, noisy perturbation or other steeper region of the
> space, our learning rate may be excessively large, causing divergence.
> Ideally, we would like our algorithm to have the maximum freedom to
> select large learning rates if it wishes, but we would suggest
> practitioners also use this defence if they encounter problems with very
> large learning rates.
>
> ## High-Dimensional Learning Rates
>
> Since submission, we have gathered evidence suggesting that the
> surprising underperformance of our HDLR settings may be due to the
> (relatively) small quantity of validation data available to learn these
> learning rates. For comparability, we used the same dataset splits for
> all algorithms (following Gal and Ghahramani (2016) for the UCI
> datasets, and using standard 60%/20%/20% split sizes otherwise).
> However, in the HDLR case, we have the same number of learning rates as
> we do model parameters. With so many hyperparameters, it makes sense for
> the validation dataset to be enlarged to accommodate their tuning: an
> excessively small dataset could stymie our optimisation efforts.
>
> We have since performed some small additional experiments to investigate
> test performance as a function of validation dataset size, considering
> *Our (WD+LR+M)* and *Our (WD+HDLR+M)* cases on UCI Energy. The results
> show both settings benefit from larger validation datasets, but the HDLR
> case more so, with average performance matching the scalar case when 25%
> of the combined training-validation data is assigned to the validation
> set, and beating the scalar case for larger validation sets (37.5% and
> 50% of the combined training-validation data). The HDLR best-case
> performance still does not beat the scalar setting, indicating there are
> additional factors at play in the high-dimensional configuration.
> However, as we have stated in our paper and in other responses, we seek
> to improve the performance of an arbitrarily selected hyperparameter
> initialisation, which motivates focussing on average performance as well
> as best-case performance, and the former does show benefits. We have
> updated our paper to include these new results (Section B.8.1).
>
> ## Other Remarks
>
> While the in-paper link to our code is redacted for anonymous
> submission, the contents of our repository are made available to
> reviewers in the supplementary material.
>
> Thank you for pointing out some incorrect references to arXiv pre-prints
> instead of peer-reviewed publications. We have amended these in our
> updated paper (References).

---

> > ### Comment · Reviewer_1R41 · 2021-11-26
> > **Upgrading score**
> >
> > Thank you for clarifying the choice of the meta-hyperparameters and the additional insights into HDLR. As this mitigates my initial concerns, I have upgraded my score.

---

### Official Review · Reviewer_QKup · 2021-10-28

**Correctness:** 4
**Technical Novelty And Significance:** 3
**Empirical Novelty And Significance:** 2
**Recommendation:** 8
**Confidence:** 4

**Main Review:**

This paper starts from (Lorraine, 2019) and that it can be generalized in a rather straightforward manner. Crucially, this generalization allows to tune optimizer parameters as well, such as learning rate and momentum. It is well known, and acknowledged by the original authors, that (Lorraine, 2019) is very sensitive to the correct choice of the step size eta, and the current work claims that by including this parameter into the optimization, these problems can be circumvented. At least in the experiments presented here, this is very convincingly the case. The paper certainly excels at fixing issues with (Lorraine, 2019). I am not so sure it adds very much on top of that.

First, while the experiments cover a good range of different setups, they are a little set up to make the proposed approach look good compared to (Lorraine, 2019). Only 3 HPs are optimized over, 2 of which are optimizer parameters which the other method cannot tune, and the authors just leave them fixed. These problems do not seem to be very hard, given that the simple baseline of training 3x with random initialisations and picking the best is very competitive on all of them. And why only go for 3 HPs if you can compute a gradient? I have comments on the experiments below.

To me, an important question is what this work adds to the HPO state of the art in general. (Lorraine, 2019) is obviously quite limited, but so is the current work and all gradient-based HPO. What do you do with integer or categorical hyperparameters? How do you make use of parallel computations (which is key in practice)? How do you support modern optimizers like Adam or AdaGrad, which are not just a simple equation like (4)? Most importantly, how do you ensure robust behaviour, and in particular avoid catastrophic failures which hypergradient approximate methods are prone to? I'd love the paper to make some statements, and in general tell the reader what is done more here than fixing (Lorraine, 2019).

For a start, I'd love a much better explanation of what you say in 2.5, which I did not fully understand, including Figure 2. What you say here seems to be key to explain the advantages over (Lorraine, 2019). Can you underline this with some analysis and experiments?

Next, is there a way to make your approach "one step more accurate"? Right now, you just do the same as (Lorraine, 2019) in terms of the assumptions (which are strong). Why not for example take w_i and w_{i-1}, then assume that derivs at w_j, j < i, are all equal to derivs at w_{i-1}, and that w_i -> w_*? Is that still tractable? Do you need new interesting tricks to make it work? This would add something to the machinery of (Lorraine, 2019).

Finally, more comments on the experiments. They do serve to show that the current approach works better than (Lorraine, 2019) and another hypergradient approximation, on a setup which mostly tunes optimizer parameters. But I feel they do not do much more. First, overall running times are missing, so it is nearly impossible to see what the results mean. Nowadays, competitive HPO experiments show curves with wall-clock time on the x axis.
More importantly, an obvious baseline is missing: population based training (PBT). This is also tuning models while training them, and is strictly a constant factor more expensive than training a model once. This factor is spent on parallel compute. PBT is implemented in Ray Tune and should work out of the box. If you claim your method costs 3x more than one training, just use 3 parallel workers. Another meaningful baseline would be ASHA (https://arxiv.org/abs/1810.05934), also available in Ray Tune. You can run it with a time cutoff of about 3x a single training run.

For gradient-based HPO, there are probably two ways forward. Either (a) show that a method can work robustly and better than something simple like asynchronous Hyperband (ASHA) or PBT (see below), none of which are even close in methodological complexity, work in general settings. Or (b), find a meaningful setup which has a lot of HPs (they need to be continuous, though), argue why it is important, and then apply gradient-based HPO, because other methodology will not apply. But if you evaluate your gradient-based approach on simple setups like the ones used here, with 3 HPs, then you really have to do it in a way that includes non-gradient-based SotA HPO methods.


**Summary Of The Paper:**

This paper is concerned with tuning continuous hyperparameters of a neural network setup (model, loss function and optimizer parameters) by way of a cheap approximation to the hypergradient. It is closely related to earlier work (Lorraine, 2019), which however overlooked that the idea can be generalized to optimizer parameters (e.g., learning rate, momentum), which is effectively done here. In a range of well-done experiments, the method is shown to very convincingly outperform (Lorraine, 2019), and in some cases even the "gold standard" approach to differentiate through the whole learning trajectory. Code is provided.


**Summary Of The Review:**

This work provides a simple extension of earlier work (Lorraine, 2019), which ends up being rather important, since the former simply does not work very well at all on standard HPO problems. However, the work does not do much more than that, It stays methodologically very close to this earlier work, and does not really address any of the shortcomings of gradient-based HPO.

The experiments are well designed to show superiority over (Lorraine, 2019) and other gradient-based HPO, but lack comparison to relevant other HPO baselines (ASHA, PBT), and results are not presented in terms of time spent, so they are hard to interpret.

---

> ### Author Response · Authors · 2021-11-22
> **Response to Reviewer QKup**
>
> Thank you for taking the time to review our submission; we really
> appreciate your efforts. We’d like to address your comments and
> questions below.
>
> ## Contribution to HPO in General (Paragraphs 2, 3 and 5)
>
> The principal objective of our algorithm is to take an arbitrary
> hyperparameter initialisation and improve the post-training performance
> which would be achieved by fixing the hyperparameters at that
> initialisation. This is orthogonal to the question of exploiting
> parallel computation to efficiently compare multiple initialisations;
> our proposal is compatible with such existing methods, but it need not
> consider anything more than the current run to achieve our objective.
> (See also our later comments regarding experiments on PBT and ASHA.)
>
> We are aware of the limitations of gradient-based hyperparameter tuning
> as a high-level approach, such as integer/categorical hyperparameter
> support and robustness, and have updated the paper to make these more
> prominent (last sentence of Section 3). However, these general
> limitations do not detract from our contributions to this sub-field. We
> argue that an effective guarantor of robustness is to compare our method
> on multiple initialisations using existing techniques: we expect our
> algorithm would improve performance of each initialisation individually,
> and considering an entire population of results defends against any
> individual trial crashing or diverging for any reason.
>
> In Equation 4 (and throughout our derivation), we notate $\mathbf{u}$
> as depending only on the current weights $\mathbf{w}$ and
> hyperparameters $\boldsymbol{\lambda}$ for clarity. By this notation, we
> mean for $\mathbf{u}$ to refer to any iterative optimisation algorithm
> which uses the current $\mathbf{w}$ and $\boldsymbol{\lambda}$ to
> compute the necessary weight update; our derivations remain valid for a
> variety of optimisers. In particular, there is no difficulty if
> $\mathbf{u}$ depends on some internal state/memory/buffer (although an
> implementation would need to truncate the gradient history of that
> state/memory/buffer to avoid the memory requirement growing unbounded,
> which represents an additional approximation). Thus, our algorithm
> trivially supports Adam, AdaGrad and a whole host of other
> gradient-based optimisers.
>
> Finally, there are several ways our approach could be extended to be
> “one step more accurate” as you describe, at the cost of larger
> computational overhead, but we argue these investigations would worthily
> be the subject of future, independent work. We believe our extension of
> Lorraine et al.’s method to a much broader class of hyperparameters, and
> the associated interpretation of optimiser hyperparameters directly
> modifying the loss surface (see below), represent a cohesive
> contribution in their own right.
>
> _(continued...)_

---

> > ### Author Response · Authors · 2021-11-22
> > **Response to Reviewer QKup (continued)**
> >
> > _(continued from above)_
> >
> > ## Intuitive Interpretation in Section 2.5 (Paragraph 4)
> >
> > Lorraine et al. (2019) note a methodological restriction on their
> > development, which prevents them applying their algorithm to optimiser
> > hyperparameters (e.g. learning rate and momentum). They base their
> > derivations on the training loss $\mathcal{L}\_{T}$, which does not
> > vary with learning rate and momentum. Thus, differentiating
> > $\mathcal{L}\_{T}$ with respect to the hyperparameters yields no
> > information useful for updating these optimiser hyperparameters.
> > Further, the use of the Implicit Function Theorem to motivate their
> > approximation requires $\mathcal{L}\_{T}$ (or at least
> > $\frac{\partial\mathcal{L}\_{T}}{\partial\mathbf{w}}$) to be an
> > implicit function of all the hyperparameters, which is clearly not the
> > case.
> >
> > In contrast, we do not work with $\mathcal{L}\_{T}$, but with an
> > arbitrary update $\mathbf{u}$, which then does depend on the optimiser
> > hyperparameters. However, the optimisation objective remains the same,
> > so this does not resolve the above methodological concerns raised by
> > Lorraine et al. To address these, we look to recast our development in
> > the original form of Lorraine et al. We do this by defining some
> > ‘pseudo-loss’ $\overline{\mathcal{L}}$ such that our weight updates
> > are equivalent to
> > $\mathbf{w \leftarrow w} - \frac{\partial\overline{\mathcal{L}}}{\partial\mathbf{w}}$.
> > Acknowledging that our objective condition is now zero-update rather
> > than zero-derivative-of-$\mathcal{L}\_{T}$, we now have the functional
> > dependencies necessary for the original argument of Lorraine et al. to
> > hold. The general expression for $\overline{\mathcal{L}}$ for any
> > particular problem is
> > $\overline{\mathcal{L}} = \int_{}^{}{\mathbf{u}\ \mathrm{d}\mathbf{w}}$,
> > by comparison with Equation 4.
> >
> > This mathematical construction has a geometric interpretation in simple
> > cases. If we consider the learning rate $\eta$ to be our only
> > hyperparameter, we have
> > $\overline{\mathcal{L}} = \eta\mathcal{L}\_{T}$, which suggests two
> > equivalent interpretations of the optimisation problem we are solving.
> > In the first (conventional) interpretation, we seek the minimum of a
> > fixed loss surface $\mathcal{L}\_{T}$, so optimising $\eta$ means
> > standing at our current point on the loss surface and considering the
> > effect of taking different-sized steps across it. Clearly, this
> > interpretation decouples $\eta$ from the underlying function
> > $\mathcal{L}\_{T}$, so suffers from Lorraine et al.’s methodological
> > issue. But in the second (our novel) interpretation, we seek the minimum
> > of the loss surface $\overline{\mathcal{L}}$, but may only update our
> > position by taking a step with magnitude equal to the local gradient of
> > the surface. Now, when we optimise $\eta$, we *rescale* the surface
> > beneath the update step, and consequently also rescale the update step
> > size. In this way, we have coupled $\eta$ and $\mathcal{L}_{T}$ into
> > one objective $\overline{\mathcal{L}}$, which varies with $\eta$ and
> > so adheres to Lorraine et al.’s methodological restrictions. The key
> > idea is thus that we have reinterpreted the problem to allow optimiser
> > hyperparameters to dynamically transform the objective surface, thus
> > legitimising our derivation.
> >
> > The remainder of Section 2.5 attempts to make a similar geometric case
> > for the momentum. This is a much more intricate setting, without an
> > immediately obvious closed form for $\overline{\mathcal{L}}$, so the
> > exposition is necessarily vaguer. Suppose we are descending a
> > steep-sided valley of the loss surface using SGD with momentum. The
> > momentum acts to ‘pull’ the update step in the direction it has
> > accumulated (down the valley), so gradients aligned with the valley’s
> > downhill direction are augmented, and gradients in other directions
> > diminished. When we work with the pseudo-loss
> > $\overline{\mathcal{L}}$, the update step is fixed, and we must
> > transform the underlying loss surface to achieve the same effect. So if
> > the update step is aligned with the accumulated momentum, we must squash
> > the loss surface about our current point in this direction, such that
> > the update step lands on a new point further down the valley in the
> > original unscaled space. Similarly, if the update step is misaligned
> > with the accumulated momentum, this squashing in the momentum’s
> > direction must be combined with a stretching in the update step
> > direction, so that we make *less* progress in that direction of the
> > original space. In aggregate, this is a complex and intricate
> > transformation of the space such that the ‘correct’ new points are
> > brought to lie under the update step. This is what we mean by our
> > discussion of squashing and stretching the space depending on the local
> > gradients in Sec. 2.5. However, the general principle of transforming
> > the loss surface to fit the effect of the optimiser is more important to
> > understand than the complicated momentum example specifically.
> >
> > We have added Appendix A.3 to the paper, containing this elaboration on
> > Sec. 2.5.
> >
> > _(continued...)_

---

> > > ### Author Response · Authors · 2021-11-22
> > > **Response to Reviewer QKup (continued)**
> > >
> > > _(continued from above)_
> > >
> > > ## Experiments (Paragraphs 1, 6 and 7)
> > >
> > > Our experiments aimed to isolate the benefits provided by our specific
> > > contribution – extending the method of Lorraine et al. (2019). The
> > > clearest way of doing so was to consider methods with and without
> > > commonly-used optimiser hyperparameters (learning rate and momentum). We
> > > do also include experiments with large numbers of learning rates (the
> > > *Ours (WD+HDLR+M)* settings), noting Lorraine et al.’s previous
> > > demonstration of the effectiveness of independent weight decay
> > > coefficients for each model parameter – optimising this many
> > > hyperparameters essentially requires a gradient-based approach.
> > >
> > > **We do consider the runtimes of our algorithm.** Figure 5 shows runtime
> > > statistics and the evolution of test performance with time for
> > > Fashion-MNIST, with extensions in Figures 11 and 14 in the Appendix.
> > > Figures 16, 17 and 18 also illustrate runtimes in some of our additional
> > > experiments, as does the new Figure 19 (see below). The results show our
> > > methods to outperform their gradient-based competition and to reach the
> > > vicinity of Bayesian Optimisation’s final performance in an order of
> > > magnitude less time.
> > >
> > > As discussed above, the exploitation of parallelised computation in HPO
> > > is orthogonal to our presentation here. However, our algorithm can be
> > > used in combination with such methods, including PBT and ASHA as you
> > > mention. Both PBT and ASHA treat the training procedure as an arbitrary
> > > algorithm, which reports results and can be interrupted at specified
> > > intervals: in the vanilla case, that arbitrary internal algorithm is
> > > equivalent to our *Random* setting, but it could trivially be made one
> > > of *Our* settings instead. Our original results indicate that, in the
> > > latter setting, our algorithm would allow each trial initialisation to
> > > achieve lower losses in less time, thus complementing the macroscopic
> > > strategies employed by PBT and ASHA. **We have performed additional
> > > experiments as you suggest, and the results validate this claim:
> > > combining our algorithm with PBT or ASHA substantially improves the
> > > quality of the resulting population of fully-trained models, without
> > > significantly harming the best losses found by these population search
> > > strategies. Our paper** **has been updated to include these new results
> > > (Appendix B.8.2)**.
> > >
> > > We would also draw attention to a new Figure 19 in our Appendix, which
> > > plots the time evolution of incumbent best test loss found by PBT and
> > > ASHA, both vanilla and when combined with our algorithm. While the
> > > performance/runtime trade-off for PBT is largely unchanged between the
> > > two cases, in ASHA the higher-quality model population produced by
> > > incorporating our algorithm leads to the discovery of smaller test
> > > losses in substantially lower runtimes. Appendix B.8.2 (final paragraph)
> > > of the updated paper discusses this in more detail.

---

> ### Comment · Reviewer_QKup · 2021-11-23
> **Response to author feedback**
>
> I'd like to thank the authors for their useful feedback on my review. It made me look at the appendix more thoroughly. I especially appreciate the additional comparisons with PBT and ASHA.
>
> I will increase my score.
>
> It is just disappointing that the one use case HDLR which would benefit from gradients, does not seem to. Maybe the authors could attempt some more parameter tying (per layer, or per activation). Also, it seems that the combination with ASHA could be promising even if this method here only shines on a small number of optimizer parameters: the random selection of ASHA could work over remaining ones.

---

### Official Review · Reviewer_c7so · 2021-11-01

**Correctness:** 3
**Technical Novelty And Significance:** 2
**Empirical Novelty And Significance:** 3
**Recommendation:** 6
**Confidence:** 3

**Main Review:**

Comments:
1- "Many machine learning methods are governed by hyperparameters: quantities which, unlike model parameters or weights, control the training process itself (e.g. optimiser settings, dropout probabilities)"
I am not sure I fully agree with this sentence. I am not an expert, but from what I understand, the dropout parameters have a direct influence on the model: different dropout probabilities leads to different models, whereas different learning rates should yield the same results (with different time of convergence).
Do authors agree on this? Or did I miss something?

2- In my opinion part 2.1, 2.2, and 2.3 and too handy, the theoretical result should be encapsulate in one proposition.
What do authors mean by the approximation sign in (7)?

3- I would say that the proposed algorithm is a "one-pass" algorithm.
In algorithm 1 there are somehow hidden nested for loops with the while + for t in 1 to T. It seems that authors are not proposing a "one-pass" algorithm, but more a warm-restart algorithm.
Authors themselves wrote: "We emphasise training is not reset after each hyperparameter update — we simply continue training from where we left off, using the new hyperparameters. Consequently, Algorithm 1 avoids the time cost of multiple training restarts".
Do authors consider changing the name of the paper? Maybe something in the idea of "Implicit differentiation to jointly tune model-based and non model-based hyperparameters" ?

4- Figure shows that the learning rate seems to converge toward a value around $1O^-1$, but the weight decay does not seem to converge. Do you think that the weighted decay is not converging? Or may the stepsize is too large for the weight decay? Could authors comment on that?

5- Figure 4.b is extremely interesting. Just a question to be sure, it is written 400 hyperparameter updates, does it mean that the time budget for each "square" is different?

6- In the experiments I would like to see the exact formula of the optimization problem which is solved (either in the main paper or in appendix). In my opinion this would make the paper clearer.

7- What does "myopic convergence to the local minima" means?
I a more general way, I my opinion, authors make a lot of statements that could be described as "hand wavy", for instance
"Similarly, we suppose a ‘momentum’ could, at every point, locally *squash the loss surface in the negative-gradient direction* and stretch it in the positive-gradient direction.
In aggregate, these transformations straighten out optimisation trajectories and *bring local optima closer to the current point*."
The word "interpret" is present 34 times in the paper: this contrasts with the lack of theoretical results.


**Summary Of The Paper:**

Authors propose an extension of Lorraine 2019 to handle hyperparameters that control the convergence speed of the model: Lorraine 2019 "only" handled hyperparameters that were part of the model.

The extension seems a little incremental, but the paper is well written (except for the theoretical part), code is provided, the bibliography is quite extensive, and experiments show interesting improvements.



**Summary Of The Review:**

In conclusion I vote for week rejection because the theoretical part is too hand wavy and IMO the paper cannot be published right now. In particular, IMO, the name of the paper should be changed.
However, except this part, the paper is well written, and experiments are very interesting. If authors take into account the comments, I will raise my score.

---

> ### Author Response · Authors · 2021-11-22
> **Response to Reviewer c7s0**
>
> Thank you for taking the time to review our submission; we really
> appreciate your efforts. We’d like to address your questions below.
>
> ## ‘One-Pass’ vs ‘Warm-Restart’ Algorithm (Q3)
>
> We strongly disagree that our algorithm is a warm restarting algorithm.
> At a high level, Algorithm 1 has the following structure:
>
>   - While some exit condition is not reached:
>
>       - Perform $T$ gradient-based updates to the weights
>         $\mathbf{w}$
>
>       - Perform one gradient-based update to the hyperparameters
>         $\boldsymbol{\lambda}$
>
> Our proposal is to take a classical $N$-step training process and
> insert hyperparameter updates every $T < N$ steps. The outer \`while\`
> loop only facilitates this structure of alternating steps – there is no
> resetting or state change at the end of every loop. If we removed the
> hyperparameter update step, the algorithm would be identical to
> classical training, even though there are nested loops. For this reason,
> we describe our algorithm as ‘one-pass’, because we perform exactly as
> much weight training as in the classical case. The terminology
> ‘warm-restart’ suggests we revisit earlier weight learning steps with
> our updated hyperparameters, which is not the case.
>
> ## Precision of Commentary (Q7)
>
> By “myopic convergence to local optima”, we mean that short-horizon bias
> causes us to take hyperparameter steps which consider only short
> distances into the future (a ‘myopic’ view). Such an update strategy
> will likely cause us to seek out the nearest minimum of loss space.
> However, if this is a poor local minimum, it would be preferable to
> retain a larger learning rate, ‘jump over’ the nearest minimum and
> instead target a different, more globally optimal minimum of the space.
> A long-horizon method can (theoretically) access the latter possibility
> by considering the longer-term optimisation trajectory, but a
> short-horizon method cannot. In this sense, converging naïvely to the
> nearest local minimum is undesirable.
>
> We acknowledge our wording is more intuitive than technical in places,
> and will be glad to make changes where necessary to make our claims more
> precise and rigorous. However, we dispute that the word “interpret” is
> used 34 times in our paper. In the worst case, we have used
> “reinterpret” and its derivatives 6 times, of which three are in
> connection with Section 2.5, which seeks to provide qualitative
> intuition for our algebraic argument.
>
> In Section 2.5, we relate to the issue described by Lorraine et al.
> (2019): since hyperparameters such as learning rates and momenta do not
> alter the loss surface, it is not possible to update them by
> differentiating the loss function with respect to the same. We seek to
> demonstrate that our method, which does support these hyperparameters,
> does not methodologically conflict with that fact. To do so, we note
> that a pseudo-loss function can be defined such that it *is* altered by
> changing the aforementioned hyperparameters. This is an idea we feel is
> intuitively appealing in general, though for many specific cases the
> lack of a closed-form expression for $\overline{\mathcal{L}}$ makes an
> intuitive description difficult. Our response to reviewer QKup contains
> an elaboration on this section which may be of interest, as does
> Appendix A.3 of our updated paper.
>
> _(continued...)_

---

> > ### Author Response · Authors · 2021-11-22
> > **Response to Reviewer c7s0 (continued)**
> >
> > _(continued from above)_
> >
> > ## Other Points
> >
> > **1)**  We didn’t intend for this statement to profess a particular
> >     philosophy on the difference between parameters and hyperparameters.
> >     In a single training run, dropout probabilities would typically be
> >     fixed, and they would be tuned in a methodologically similar way to
> >     learning rates etc., so are considered at HPO time rather than
> >     training time. Similarly, quantities describing the architecture of
> >     a model would be described as hyperparameters rather than parameters
> >     in typical frameworks, although we grant they (and dropout
> >     probabilities) are more intertwined with the model than most. We
> >     have refine this wording in our updated paper (still in the first
> >     sentence of Section 1).
> >
> > **2)**  Our theoretical presentation is very much a motivational argument
> >     rather than a formal proof, so we have deliberately avoided styling
> >     it as such. Equation 7 is obtained by taking the second approximate
> >     equality of Equation 6, taking the limit $i \rightarrow \infty$ of
> >     each side, and claiming that if the LHS and RHS are approximately
> >     equal in Equation 6, then so should their limits be. There is no new
> >     approximation introduced here.
> >
> > **4)**  (We assume you refer to Figure 3 – please correct us if not.) Our
> >     fixed-hyperparameter baseline results, illustrated by the background
> >     shading, suggest performance in this setting is harmed by
> >     excessively large weight decays (above around $10^{- 2}$), but is
> >     otherwise insensitive to the weight decay. We believe this is why
> >     our trajectories do not converge to a single weight decay: many
> >     different values produce good results, so it does not significantly
> >     affect performance.
> >
> > **5)**  You are correct – we use a constant number of hyperparameter
> >     updates, which means settings with larger $T$ (more weight updates
> >     per hyperparameter update) have larger computational budgets. Either
> >     the number of hyperparameter updates or the computational budget
> >     must vary for this analysis.
> >
> > **6)**  The implementation of our experiments follows Algorithm 1 in order
> >     to solve the problem described in Equation 1, with loss functions as
> >     specified for each experiment (generally MSE or cross-entropy). Our
> >     update function $\mathbf{u}(\boldsymbol{\lambda},\ \mathbf{w})$
> >     represents PyTorch’s implementation of SGD, that is
> >     $\mathbf{u}\left( \boldsymbol{\lambda,\ w} \right) = \eta\mathbf{v}$
> >     and
> >     $\mathbf{v} \leftarrow \mu\mathbf{v +}\frac{\partial\mathcal{L}_{T}}{\partial\mathbf{w}} + \xi\mathbf{w}$,
> >     with learning rate $\eta$, momentum coefficient $\mu$ and weight
> >     decay coefficient $\xi$. Although our use of the momentum buffer
> >     $\mathbf{v}$ is not notated as a dependency of $\mathbf{u}$, the
> >     only extra care required is to detach $\mathbf{v}$ from the
> >     computational graph after each hyperparameter update.
> >
> > After computing the approximate hypergradient
> >     $\left\lbrack \frac{\partial\mathcal{L}\_{V}}{\partial\boldsymbol{\lambda}} \right\rbrack_{\boldsymbol{\lambda},\ \mathbf{w}} + \mathbf{g}\_{\mathrm{\text{indirect}}}$,
> >     we place this in the gradient fields of the hyperparameters
> >     $\boldsymbol{\lambda}$, before calling a standard gradient-based
> >     optimiser on the hyperparameters. In Algorithm 1 we show a simple
> >     gradient descent update to $\boldsymbol{\lambda}$; we have updated the
> >     paper to emphasise that this is an illustration, rather than a
> >     mandatory choice of hyper-optimisation algorithm (comment on the
> >     last line of Algorithm 1, Figure 4). Our experiments use Adam in
> >     place of this simple update, so our actual hyperparameter update
> >     follows the pseudocode of Adam with
> >     $\left\lbrack \frac{\partial\mathcal{L}\_{V}}{\partial\boldsymbol{\lambda}} \right\rbrack_{\boldsymbol{\lambda},\ \mathbf{w}} + \mathbf{g}\_{\mathrm{\text{indirect}}}$
> >     in place of the gradient of the optimisation objective ($g_{i}$ in
> >     the original Adam paper).

---

> > ### Comment · Reviewer_c7so · 2021-11-28
> > **Re: ‘One-Pass’ vs ‘Warm-Restart’ Algorithm**
> >
> > I understand author's point: authors mean one pass to compute the hypergradient.
> > I was worried this could be misleading to the reader because the hypergradient computation is nested into the outer, leading to a global algorithm which is not 'one-pass' (while the computation of the hypergradient is).
> > I just wanted the authors to be aware of the potential misunderstanding, given the authors response I think my remark becomes a minor problem.

---

### Official Review · Reviewer_iZEW · 2021-11-03

**Correctness:** 4
**Technical Novelty And Significance:** 3
**Empirical Novelty And Significance:** 3
**Recommendation:** 8
**Confidence:** 4

**Main Review:**

**Strengths:**

- The technical deductions look correct (although I did not check every line).
- The authors are candid about the extent of their contribution and properly cited relevant literature.
- The writing is overall clear. A number of clarifications and illustration figures were provided.
- There are abundant experiments, with many comparisons among different methods and ablation studies of the proposed method.

**Weaknesses, and questions to authors:**

Despite the above strengths, there are some vague parts in the current version. I would raise my score if the authors could properly address the questions, and would suggest the authors incorporate them into the revision.

- As (honestly) claimed in Section C.4, the technical advancement from Lorraine et al. (2019) seems incremental: In the framework of Lorraine et al. (2019), $\mathcal{L}_T(\lambda, w)$ could still be parameterized by optimization hyperparameters in the framework of the previous work, so that $u(\lambda, w)$ could be expressed as a closed form.
The authors may want to provide more details around Algorithm 1 or Section 2.5 on how $u(\lambda, w)$ is computed, and show a table of  examples on simple tasks (e.g., SGD on ridge regression) to clarify what $u(\lambda, w)$ would look like, so that we can see why it is different from $\eta \partial \mathcal{L}_T (\lambda, w)/ \partial w$.
- The mean and median performance in Table 1 and 2 may not be quite meaningful in the hyperparameter tuning context: for the methods that do not optimize over optimization hyperparameters, “Best” instead of “Mean” or “Median” may be more interesting to practitioners. The 200 random initializations can be regarded as candidates selected in random search of hyperparameters, and users would want to pick the one with the smallest error (Of course, the small variance is still a plus). Are there examples of the best models (whose performance are in the “Best” column) found by different methods, so that readers can see how different they are? And, is the proposed method stable in terms of finding more similar models from different initializations? This type of information helps to show that learning rate tuning really matters on these datasets, since the best test MSEs are often achieved by other methods.
- Why would a large $i$ be a problem for Algorithm 1 (as described in Section 4.3), when the weight $w$ is fixed in the “for j in 1 to i” loop? In fact, the first approximately equal sign in Equation 6 would become an equal sign here.

Also, the link to the source code points to an empty GitHub account.

**Summary Of The Paper:**

This paper tackles the hyperparameter optimization problem with a one-pass approach that alternatively optimizes over machine learning model parameters and hyperparameters. To optimize over a bilevel optimization problem, this paper generalizes Lorraine et al. (2019) by replacing the gradient update of parameters with a more general parametric form, thus allowing the hyperparameters to be updated alongside parameters by gradients coming from truncated Neumann series.

**Summary Of The Review:**

This paper overall does a good job in presenting the technical contribution with deductions, pseudocode and experiments. Ambiguities do exist; I would encourage the authors to address them, so as to make the paper more readable.

********* Updates after the discussion period *********

This paper is a good paper. I have increased my score.

---

> ### Author Response · Authors · 2021-11-22
> **Response to Reviewer iZEW**
>
> Thank you for taking the time to review our submission; we really
> appreciate your efforts. We’d like to address your questions below, in
> the order you present them.
>
> ## Advancement over Lorraine et al. (2019)
>
> Fundamentally, Lorraine et al.’s (2019) formulation requires us to work with the training loss $\mathcal{L}\_{T}$, on which all hyperparameters
> must depend, while we work with an arbitrary update $\mathbf{u}$. As
> you note, in the case of vanilla SGD (without weight decay or momentum),
> this difference is purely notational: although the loss function does
> not depend on a learning rate, if we define
> $\overline{\mathcal{L}} = \eta\mathcal{L}_{T}$, then a trivial
> adjustment to Lorraine et al.’s derivation allows their method to be
> applied to $\overline{\mathcal{L}}$ with minimal difficulty. But this
> only works because we can easily interpret the learning rate $\eta$ as
> modifying some ‘loss function’ $\overline{\mathcal{L}}$ without
> dramatically changing our problem.
>
> If instead we were to consider SGD with momentum, it is much harder to
> construct an $\overline{\mathcal{L}}$ which depends on the momentum
> factor and gives the weight update through
> $\frac{\partial\overline{\mathcal{L}}}{\partial\mathbf{w}}$ alone.
> This is where we diverge from Lorraine et al.: although such an
> $\overline{\mathcal{L}}$ theoretically exists, it is extremely
> inconvenient to calculate, and requires a manual derivation for every
> optimiser and combination of hyperparameters. By contrast, it is much
> easier to incorporate the momentum into a differentiable weight update
> $\mathbf{u}$, which then additionally depends on some hidden momentum
> buffer which we do not notate for clarity. (Our response to reviewer
> QKup contains an elaboration on this matter which may be of interest.)
> Further, that weight update is readily available during implementation:
> it is whatever object we add to/subtract from the network weights at the
> end of our optimiser code. Thus, our reformulation makes the application
> of Lorraine et al.’s method to optimiser hyperparameters much more
> accessible, and provides a viable route to implementation. This extends
> even to optimisers other than SGD, for instance Adam.
>
> ## Performance Metrics in Experimental Results
>
> The purpose of our experiments is not to obtain the single best test
> performance from a variety of initialisations; clearly, if *Random* were
> to stumble upon the optimal hyperparameters in one of its sampled
> initialisations, we would not expect to be able to beat that performance
> with our method. Instead, we seek to demonstrate robustness to a variety
> of initialisations – ideally, we should be able to take arbitrary
> initial hyperparameters, and adapt these over the course of training to
> achieve good final performance. Thus, we focus our comparison on the
> average final performances across all initialisations, which demonstrate
> a substantial stability gain over random search. For that reason, we do
> not focus on the best results, except to note that our methods’ average
> performance suggests they drive many of their initialisations to final
> losses which are very similar to these best-case figures.
>
> ## Problems with Large Look-Back Distance
>
> The problem with a large $i$ in Algorithm 1 is not one of computation
> or memory; as you correctly note, the fact that $\mathbf{w}$ is held
> constant makes the impact minimal. Instead, the problem is to do with
> the accuracy of the approximation.
>
> In the first approximate equality of Equation 6, we replace the sequence
> $\mathbf{w}\_{0},\mathbf{w}\_{1},\cdots,\mathbf{w}\_{i - 1}$ from
> Equation 5 with $\mathbf{w}\_{i}$ at all time steps. (Note this means
> we cannot make the first approximate equality in Equation 6 exact by
> simply increasing $i$.) Because the $j$th summation step of Equation
> 5 involves a product of terms in $\mathbf{w}\_{i - 1}$ back to
> $\mathbf{w}\_{i - 1 - j}$, the approximation of each summation step in
> Equation 6 becomes less and less accurate as $j$ increases, as we
> expect earlier model weights to become more different from
> $\mathbf{w}_{i}$. A large $i$ causes us to continue the summation up
> to larger $j$, so it introduces progressively less accurate terms into
> our approximations. We hypothesise this is why we observe poor
> performance from larger $i$ in the experiments of Section 4.3.
>
> ## Source Code
>
> Although the GitHub link in our paper is redacted for anonymous
> submission, the contents of our repository are provided in the
> Supplementary Material.

---

> > ### Comment · Reviewer_iZEW · 2021-11-28
> > **Thanks, and further questions**
> >
> > I appreciate the authors for the detailed response. I have some further questions/comments as below:
> >
> > > Advancement over Lorraine et al. (2019), the non-SGD case
> >
> > Thanks for the explanation; indeed, the non-trivial difference of the proposed method to Lorraine et al. (2019) is in the non-SGD case. I wonder: in such non-SGD cases, how should Algorithm 1 be implemented (or to be more specific, what is the $u(\lambda, w)$ term throughout Algorithm 1)?
> >
> > The current manuscript can be clearer on that: up to Page 5, the explanations were based on abstract $u(\lambda, w)$ and $\bar{\mathcal{L}}$, and only mentions the technically trivial $u_\text{SGD}$ now and then. While in experiments, Adam is used instead of SGD. What is $u_\text{Adam}$ or $u_\text{momentum}$? For this paper to be more digestible and easier to implement, it would be essential to either show a parametric form of it, or show how to approximately compute non-SGD $u(\lambda, w)$ functions in pseudocode.
> >
> > > Problem with Large Look-Back Distance
> >
> > Thanks for the explanation. Now it seems the look-back distance can’t be either too large (so that replace the sequence $w_0$, $w_1$, …, $w_{i-1}$ with $w_i$ will be too inaccurate) or too small (so that there isn’t enough terms in the truncated Neumann series). If this is correct, is there a resource-efficient way to pick a good look-back distance (which is an important hyper-hyperparameter in the proposed method)?

---

> > > ### Author Response · Authors · 2021-11-29
> > > **Second Response to Reviewer iZEW**
> > >
> > >
> > > Thank you for your further comments on our work. We outline below our
> > > responses to your additional questions.
> > >
> > > ## Non-SGD Update Functions
> > >
> > > We intend to convey that the update function
> > > $\mathbf{u}\left( \boldsymbol{\lambda},\ \mathbf{w} \right)$ is exactly
> > > the quantity which is added to the network weights $\mathbf{w}$ during
> > > an update step. For instance, in the PyTorch framework, all the
> > > pre-implemented optimisers compute some update tensor
> > > $\Delta\mathbf{w}$, then use a final line
> > > $\mathbf{w} \leftarrow \mathbf{w} - \Delta\mathbf{w}$ to actually
> > > change the weights. When automatic differentiation is used in this
> > > setup, that quantity $\Delta\mathbf{w}$ can be computed exactly as
> > > normal, then differentiating it with respect to either $\mathbf{w}$ or
> > > $\boldsymbol{\lambda}$ gives the derivatives of $\mathbf{u}$ which are
> > > required. For this reason, we hoped the definition of $\mathbf{u}$
> > > would be clear even without specifying the minutiae.
> > >
> > > As we have mentioned in the other responses (c7so, “Other Points”, \#6;
> > > QKup, “Contribution to HPO in General”, paragraph 3), in our experiments
> > > (SGD with weight decay and momentum), $\mathbf{u}$ represents
> > > PyTorch’s implementation of SGD:
> > > $\mathbf{u}\_{\mathrm{m}\mathrm{\text{omentum}}} = \eta\mathbf{v}$
> > > where
> > > $\mathbf{v} \leftarrow \mu\mathbf{v} + \frac{\partial\mathcal{L}\_{T}}{\partial\mathbf{w}} + \xi\mathbf{w}$,
> > > with learning rate $\eta$, momentum coefficient $\mu$ and weight
> > > decay coefficient $\xi$. Our derivations do not notate the dependence
> > > of $\mathbf{u}$ on the momentum buffer $\mathbf{v}$, but the
> > > existence of $\mathbf{v}$ does not invalidate our development – the only extra care
> > > required is to detach $\mathbf{v}$ from the computational graph after
> > > each hyperparameter update. Similarly, for Adam, we may read the
> > > definition of $\mathbf{u}$ straight off Adam’s definition in its
> > > original paper (Kingma and Ba, ICLR 2015): it is (in that paper’s
> > > notation)
> > > $\mathbf{u}\_{\mathrm{\text{Adam}}} = \alpha \cdot \frac{{\widehat{m}}\_{t}}{\sqrt{{\widehat{v}}\_{t}} + \epsilon}$,
> > > with $t$ indexing time and ${\widehat{m}}\_{t},\ {\widehat{v}}\_{t}$
> > > being internal states. The latter depend directly on the additional
> > > states $m\_{t},\ v\_{t}$, and ultimately on the gradient
> > > $\nabla_{\theta}f_{t}\left( \theta_{t - 1} \right)$ (which is
> > > $\frac{\partial\mathcal{L}_{T}}{\partial\mathbf{w}}$ in our notation).
> > > Again, other than detaching these internal states from the computational
> > > graph periodically, there is no adjustment required to our algorithm to
> > > support this more complex optimiser.
> > >
> > > Finally, we emphasise that, in our experiments, Adam is used as the
> > > *outer* optimiser over the hyperparameters $\boldsymbol{\lambda}$ only.
> > > The update function $\mathbf{u}$ relates to the *inner* optimiser over
> > > the weights $\mathbf{w}$, which only ever uses SGD (with weight decay
> > > and momentum) in our empirical work. Thus, while our framework supports
> > > the use of Adam as the inner, network weight optimiser (with a
> > > corresponding $\mathbf{u}_{\mathrm{\text{Adam}}}$), we do not explore
> > > that choice in our work.
> > >
> > > ## Look-Back Distance
> > >
> > > We agree: there is a trade-off to be struck between large and small
> > > look-back distances, for exactly the reasons you describe. Our
> > > investigations of this effect are the subject of Appendix B.2.1 and
> > > Figures 4b and 6. The comparison is over-constrained; we cannot fix the
> > > computational budget of each configuration without changing the number
> > > of hyperparameter updates performed, which skews our results. That said,
> > > they seem to support our choices of update interval $T = 10$ and
> > > look-back distance $i = 5$, as used by Lorraine et al. (2019). We
> > > appeal also to the previous work we cite on p6 (Franceschi et al. 2017,
> > > Franceschi et al. 2018 and Majumder et al. 2019) which, as we discussed
> > > in the response to Reviewer 1R41 (“Meta-Hyperparameters and Clipping”,
> > > paragraph 1), suggests our setting is progressively less sensitive to
> > > higher-order hyperparameters. As with most hyperparameter optimisation
> > > techniques, the principled selection of meta-hyperparameters is hampered
> > > by the computational costs of repeated experiments, which we do not
> > > explore.

---

> > > > ### Comment · Reviewer_iZEW · 2021-11-29
> > > > **Thanks for the answers**
> > > >
> > > > Thanks for the additional explanations. This is a good paper. I have increased my score.
> > > >
> > > > - Non-SGD Update Functions: Please consider adding the parametric forms of $u_\text{momentum}$ and $u_\text{Adam}$ and the corresponding explanations made in the discussions to the paper. This would greatly help readers understand the non-notational contribution and make Algorithm 1 easier to implement.
> > > >
> > > > - Look-Back distance: Thanks for the pointers. Please consider adding the discussion on counteracting effects of Neumann series truncation and $w$ approximation to Section 4.3.

---

### Author Response · Authors · 2021-11-22
**Paper Updated (Response to All Reviewers)**

Thank you to all the reviewers for your comments. We have updated the
submitted version of our paper to reflect the following changes, which
are discussed in more detail in the corresponding individual responses:

  - c7s0 \#1: Refining our definitions of hyperparameters vs model
    parameters to make these consistent with our examples (Section 1,
    Paragraph 1)

  - c7s0 \#6: Clarifying that our use of gradient descent to update the
    hyperparameters $\boldsymbol{\lambda}$ in Algorithm 1 is illustrative,
    not mandatory; another optimiser (e.g. Adam) could be used instead
    to update the hyperparameters (Algorithm 1, Figure 4)

  - 1R41: Including results studying the effect of validation dataset
    size on _Our (WD+HDLR+M)_ case, which show it is a factor limiting
    performance in that setting. This explains the underperformance of
    _Ours (WD+HDLR+M)_ relative to _Ours (WD+LR+M)_: the former seems to
    require more validation data, probably because it must tune many
    more hyperparameters. (Appendix B.8.1)

  - 1R41: Replacing arXiv references with updated peer-reviewed
    publications (References)

  - QKup: Making the general limitations of gradient-based
    hyperparameter tuning more prominent (Section 3, last paragraph)

  - QKup: Adding an Appendix to flesh out the intuition given in Section
    2.5 (Appendix A.3)

  - QKup: Additional experiments on ASHA and PBT, including a runtime
    analysis (Appendix B.8.2)

In response to comments by reviewer QKup, we also emphasise that runtime
results for our experiments are shown in Figures 5, 11, 14, 16, 17, 18
and 19 (new).

---

### Decision · Program_Chairs · 2022-01-20

**Decision:**

Accept (Spotlight)

**Comment:**

The paper provides a method for with tuning continuous hyperparameters (HPs). It is closely related to a previous work (Lorraine, 2019) that was limited to certain HPs, and in particular could not be applied to HPs controlling the learning such as learning rate, momentum, and are known to be influential to the convergence and overall performance (for non-convex objectives).
The reviews indicate a uniform opinion that the paper tackles an important problem, that its methods provide a non-trivial improvement over previous techniques and in particular those of (Lorrain, 2019), and that the provided experiments are extensive and convincing. The initial reviews had several concerns about technical details in the paper such as the analysis or how the meta-hyperparameters are tuned. However, in the discussions the authors provided adequate responses, resolving these concerns. I believe that with minor edits that are possible to get done by the camera-ready deadline the authors can incorporate their responses into the paper making it a welcome addition to ICLR.